# GENERATING CAD CODE WITH VISION-LANGUAGE MODELS FOR 3D DESIGNS

**Kamel Alrashedy**,* **Pradyumna Tambwekar**\*, **Zulfiqar Zaidi,**
**Megan Langwasser**, **Wei Xu**, **Matthew Gombolay**
Georgia Institute of Technology, GA, USA
{kalrashedy3,ptambwekar3,zzaidi8,mlangwasser3}@gatech.edu
{wei.xu,matthew.gombolay}@cc.gatech.edu

## ABSTRACT

Generative AI has revolutionized the fields of Design and Manufacturing by providing efficient and automated methods for generating and modifying 3D objects. One approach involves using Large Language Models (LLMs) to generate Computer-Aided Design (CAD) scripting code, which can then be executed to render a 3D object; however, the resulting 3D object may not meet the specified requirements. Testing the correctness of CAD-generated code is challenging due to the structural intricacies of 3D objects that are not discernable in code. In this paper, we introduce CADCodeVerify, a novel approach to iteratively verify and improve the design output of 3D objects generated from CAD code. Our approach provides ameliorative feedback by prompting a Vision Language Model (VLM) to generate and answer a set of validation questions to verify the generated object and prompt the VLM to correct deviations. To evaluate CADCodeVerify, we introduce, *CADPrompt*, the first benchmark for CAD code generation, consisting of 200 natural language prompts paired with expert-annotated scripting code for 3D objects to benchmark progress. Our findings show that CADCodeVerify improves VLM performance by providing visual feedback by enhancing the structure of the 3D objects and increasing the compile rate of the compiled program. When applied to GPT-4, CADCodeVerify achieved a 7.30% reduction in Point Cloud distance and a 5.5% improvement in compile rate compared to prior work. Code and data are available at https://github.com/Kamel773/CAD_Code_Generation

## 1 INTRODUCTION

Generative AI, such as Large Language Models (LLMs) offers a unique opportunity to enhance productivity, reduce costs, and increase efficiency within Design and Manufacturing sectors (Kumar et al., 2023). These industries are critical contributors to the global economy and responsible for creating products and infrastructure. Recent research has demonstrated that Generative AI can support the generation, evaluation, and correction of 3D object designs (Nelson et al., 2023; Kodnongbua et al., 2023; Makatura et al., 2023). However, while these solutions improve efficiency for designers and engineers, they often lack effective feedback mechanisms or refinement loops to automatically address inaccuracies in the initially generated 3D object.

Our research explores the potential of using VLMs to generate and refine Computer-Aided Designs (CAD), the CAD a leading approach in industrial design for creating, modifying, analyzing, and optimizing 3D objects (Sarcar et al., 2008). Designing products with CAD software such as FreeCAD (Riegel et al., 2016) and AutoCAD (Yarwood, 2013) generally requires substantial training and domain expertise. CAD software often includes scripting languages that allow users to build parametric 3D objects using code, rather than relying solely on the user interface. Leveraging the code-generating capabilities of Generative AI enables users to bypass the complexities of traditional CAD software by directly generating the underlying scripting code for representing 3D objects.

We define the process of generating and refining 3D objects using LLMs or VLMs, through CAD scripting code, as CAD code generation. Designs produced by off-the-shelf VLMs or LLMs often

---

*These authors contributed equally to this work

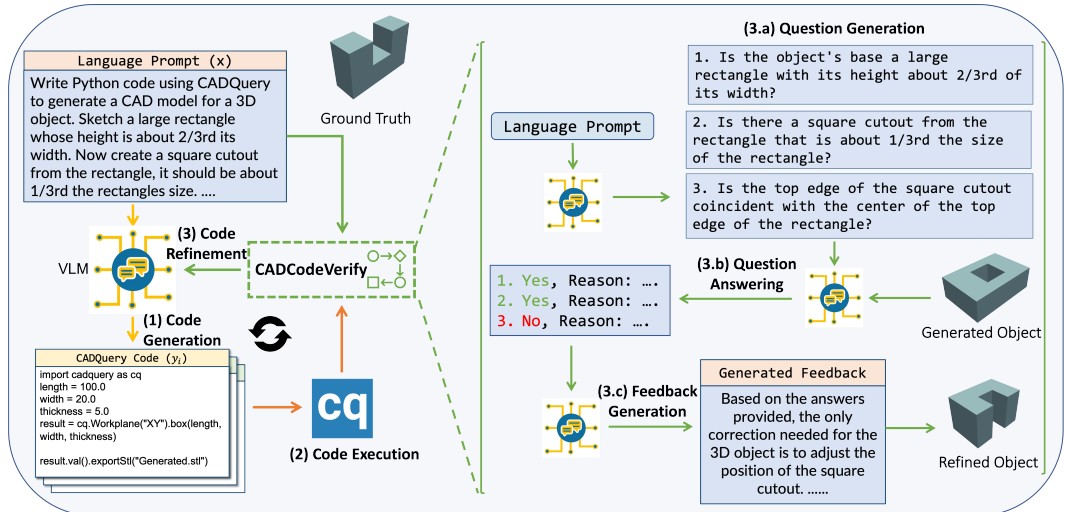

Figure 1: Our approach enables VLMs to automatically generate and refine 3D objects through a CAD scripting code (e.g., CADQuery) in three steps; (1) Code Generation, where the VLM generates CAD scripting code from a language prompt, (2) Code Execution, where the code generated by the model is rendered as a 3D object through a compiler, and (3) Code Refinement, wherein the language model engages in a self-initiated question-answering process to validate the generated object, with respect to the initial prompt, to generate actionable feedback to refine the code.

deviate functionally or structurally from stakeholder specifications as these models often hallucinate on complex, out-of-distribution tasks. Developing refinement methods to correct inaccuracies in a generated design is a critical next step in this research. Current state-of-the-art refinement methods are contingent on human-in-the-loop expertise (Makatura et al., 2023; Nelson et al., 2023), which can be time- and cost-prohibitive. Instead, we propose to develop automated feedback mechanisms for CAD code generation to reduce the barrier to entry for design and expedite lengthy design processes.

In this work, we introduce CADCodeVerify, an automated method for refining CAD code generation. CADCodeVerify eliminates the need for human involvement by generating and answering validation questions based on user requirements, offering feedback to refine 3D object code. This feedback is used to iteratively refine the design. To evaluate the performance of CADCodeVerify, we introduce a novel benchmark, *CADPrompt* contains 200 3D objects annotated with natural language prompts and expert-written Python code. This multimodal benchmark enables researchers to evaluate aspects of CAD code generation, including object quality and the syntactical correctness of the generated code. Our experimental results demonstrate that CADCodeVerify not only enhances the quality of generated 3D objects but also improves the performance of LLMs and VLMs by increasing the compile rate of compiled programs. Our work presents three key contributions:

1. We propose CADCodeVerify, a novel CAD code refinement method that enables a VLM to visually inspect generated objects and provide corrective feedback, through a question-generation and answering process, to resolve any deviations from the user's specifications.
2. With our benchmark, *CADPrompt*, we provide the first quantitative evaluation of CAD code generation across GPT-4, Gemini 1.5 Pro, and CodeLlama.
3. We demonstrate that CADCodeVerify sets a new state-of-the-art 3D design via CAD scripting, achieving a 7.30% reduction in Point Cloud distance and a 5.5% increase in successful object generation using to GPT-4, the leading VLM for CAD code generation.

## 2 RELATED WORK

### 2.1 LLMS FOR CODE GENERATION

LLMs have demonstrated impressive results in a variety of code generation applications. The most popular application is the text-to-code task, wherein users prompt LLMs with a code description,

enabling the LLM to generate the code (Rozière et al., 2023; Svyatkovskiy et al., 2020; Brown et al., 2020; Poesia et al., 2022). Another application is the text-to-SQL task, where a database and a question are provided to the LLMs, which then generate the corresponding SQL query (Rajkumar et al., 2022). Building on these advances, we explore how LLMs can support non-experts and designers in generating 3D objects via CAD code.

Prior research has demonstrated that LLMs can significantly improve the quality their generated code by incorporating feedback (Chen et al., 2023a). One such approach involves LLMs providing their own feedback by generating explanatory rationales for the initially generated code. These rationales are then used as feedback, along with the initial code, to refine the output (Chen et al., 2023b). Another approach relies on feedback from external tools, such as static code analysis (Alrashedy et al., 2024), or a Python Interpreter (Madaan et al., 2023).

In contrast, the self-correction method (Welleck et al., 2022) trains two separate models: a generator, which creates the initial output, and a corrector, which refines the generated code. In CAD code refinement, one line of research introduces a basic feedback mechanism (Yuan et al., 2024), wherein VLMs refine 3D object code based both on a script and an image. However, this method still faces challenges in terms of feedback efficiency and refinement precision. In this work, we propose a novel approach to generating visual feedback that improves the shapes, surfaces, and dimensions of 3D objects through more effective code refinement.

## 2.2 LANGUAGE FOR AUTOMATED DESIGN AND MANUFACTURING

Learning a shared embedding space between language and 3D objects is essential for integrating language into design and manufacturing. Recent advancements have introduced foundational multi-modal transformers for image-text data (Radford et al., 2021; Li et al., 2023a; Liu et al., 2023) which can be used to perform a variety of tasks. These techniques have inspired similar approaches for incorporating 3D objects into shared embedding spaces (Zeng et al., 2023; Xue et al., 2023; Yu et al., 2022; Guo et al., 2023), enabling a wide range of multimodal tasks such as dialogue, classification, and evaluation involving 3D data (Hong et al., 2023; Xu et al., 2023). While successfully learning a shared-embedding space, these approaches are not capable of generating 3D objects.

Generative modeling of 3D data has traditionally relied on probabilistic models for 3D space. These include: Generative Adversarial Networks (GANs) (Goodfellow et al., 2014), Variational AutoEncoders (VAEs) (Kingma & Welling, 2013), and Diffusion Models (Ho et al., 2020). While GANs and VAEs (Wu et al., 2015; Achlioptas et al., 2018; Gadelha et al., 2018; Yang et al., 2019; Yan et al., 2024) have been popular approaches, diffusion models have recently emerged as the state-of-the-art for probabilistic 3D modeling (Zeng et al., 2022; Koo et al., 2023). Diffusion models enable controllable object generation through language-guided shape generation and completion. However, their application in producing manufacturable designs remains limited (Li et al., 2023b; Kodnongbua et al., 2023). These models typically generate point clouds or voxels, which are non-parametric and not easily adapted for manufacturing. As such, there is a need for methods that produce parametric outputs suitable for manufacturing.

Recent research has shown that VLMs can generate designs using parametric CAD code (Makatura et al., 2023; Nelson et al., 2023). However, these approaches still require significant human feedback to produce code ensure the generated codes meets the user's specifications. In this paper, we address this limitation by developing CADCodeVerify, which autonomously generates and refines 3D objects.

## 3 CAD CODE GENERATION

As shown in Figure 1, we employ a three-step process: (1) code generation (§3.1), (2) code execution (§3.2), and (3) code refinement via CADCodeVerify for CAD code generation(§3.3).

## 3.1 CODE GENERATION

We prompt the VLM to generate initial CAD script code, $y_0$, based on a natural language description of the 3D object, $x$, along with task specifications. For the few-shot experiment, we incorporate a set of few-shot demonstrations, $E_g$, in the prompt (see Appendix B.4). We utilize CADQuery, a Python-based parametric CAD language with a built-in compiler that interprets code and renders

parametric 3D object. This CAD code generation can be broadly formulated as per Eq. 1.

$$y_0 \sim P_{LM}(y_0|x, E_g). \tag{1}$$

## 3.2 CODE EXECUTION

The generated CAD code, $y_0$, is subsequently executed by the CADQuery compiler, $\psi$, to produce the initial CAD design, $d_0$, in the Standard Triangle Language (STL) format,[1] where 0 indicates the initial version of both the generated code and CAD design. We denote this process as $d_0 = \psi(y_0)$. Occasionally, the models generate code that fails to compile, usually due to syntax errors in the Python code. We leverage insights from previous work on code repair (Chen et al., 2023b) and adapt similar techniques for CAD code generation. To resolve syntactical issues in the code, we leverage the CADQuery compiler error as feedback, $F_{err}$, for the VLM (see Appendix Figure 12, for the prompt employed in code repair). This process is repeated until a 3D object, $d_k$, is successfully rendered by $\psi(y_k)$, or the maximum number of iterations, $N$, is reached as per Eq. 2.

$$y_k \sim \{P_{LM}(y_k|x, y_{k-1}, F_{err})\}_{k=1}^{N}. \tag{2}$$

## 3.3 CODE REFINEMENT VIA CADCODEVERIFY

To address discrepancies in the generated design, we use a feedback loop to further refine the CAD code, $F_{ref}$, to further refine $y_k$ as described by Eq. 3, where $M$ is the number of refinement iterations. Optionally, we include a set of four reference images of the generated object, captured from different angles (0, 90, 180, and 270 degrees), $I_{ref} = \{I_l^0, I_l^{90}, I_l^{180}, I_l^{270}\}$.

$$y_M \sim \{P_{LM}(y_{l+1}|x, y_l, F_{ref}, I_{ref})\}_{l=N}^{N+M}. \tag{3}$$

The key innovation of CADCodeVerify is its feedback loop, which uses a VLM to generate visual feedback automatically, without human intervention or the need for external tools like geometric solvers (Makatura et al., 2023) (see §5.1). CADCodeVerify is a two-step process consisting of question-answer generation and feedback generation.

**(1) Question-Answer Generation:** CADCodeVerify first generates a set of binary "Yes/No" verification questions, Q = $\{q_1, q_2, \ldots, q_n\}$, based on the language description, $x$, and a set of few-shot example questions, $E_q$, i.e., Q $\sim P_{LM}(Q|x, E_q)$. CADCodeVerify generates between two to five questions per example (see Figure 11 in the Appendix). To answer these questions, CADCodeVerify uses the reference images of the generated 3D object, $I_{ref}$, the generated questions, $Q$, and the language description $x$. CADCodeVerify then answers each question as described in Eq. 4.

$$A \sim P_{LM}(A|x, Q, I_{ref}). \tag{4}$$

CADCodeVerify generates the answers, A = $(a_1, a_2, \ldots, a_n)$, using Chain-of-Thought (Wei et al., 2022), where each answer is accompanied by supporting reasoning (see Figure 13 in the Appendix). Furthermore, CADCodeVerify is directed to respond with "Unclear" if it determines that there is insufficient information to answer the question.

**(2) Feedback Generation:** The question-answer pairs are then used to generate ameliorative feedback to further refine the 3D object as per Eq. 5.

$$F_{ref} \sim P_{LM}(F_{ref}|Q, A). \tag{5}$$

feedback, $F_{ref}$, is then applied to refine the code, as described Eq. (3). During this step, we omit any questions for which the answer is "Yes" to allow the model to focus on addressing any unresolved issues. If all questions are answered "Yes" we assume no further refinement is necessary. The full feedback generation process is provided in Figure 14 in the Appendix. Additionally, two examples of full interactions with GPT-4 for a single iteration of CADCodeVerify are included in the Appendix ( Figures 9 and 10).

---

[1]Standard Triangle Language is also referred to as "Standard Tessellation Language". The STL (StereoLithography) file format is an openly documented format for describing the surface of an object as a triangular mesh, that is, as a representation of a 3-dimensional surface in triangular facets.

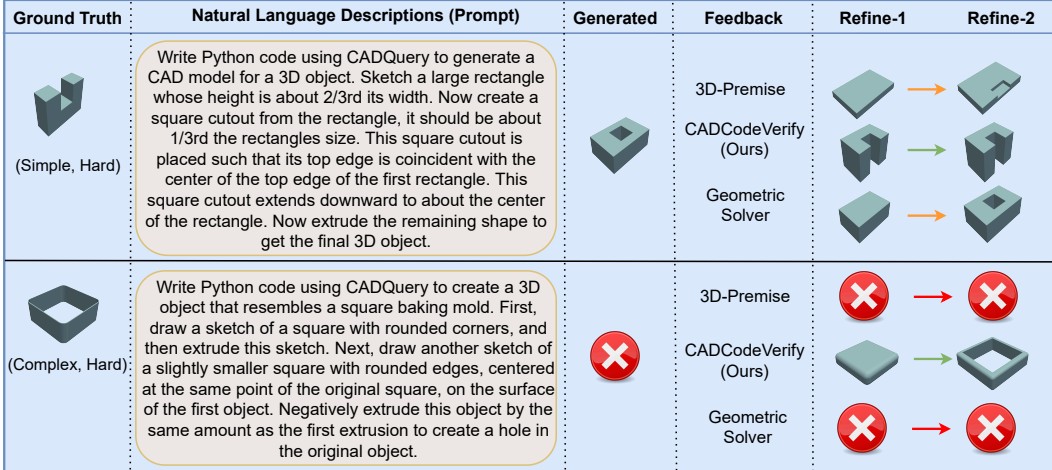

Figure 2: Examples of 3D objects generated by GPT4 after utilizing each feedback method for code refinement from our CADPrompt dataset.

Table 1: Statistics of the *CADPrompt* dataset, including the vertex and face counts of ground truth 3D objects, as well as the lengths of language descriptions and Python code (§4.3).

| Category | Structural Complexity of 3D Object | | | | | | Natural Language Descriptions | | | | | | Python Code | | | | | | Total |
| | Vertices | | | Faces | | | Words | | | Sentences | | | Lines of Code | | | Tokens | | | Datapoints |
| | min | max | avg. | min | max | avg. | min | max | avg. | min | max | avg. | min | max | avg. | min | max | avg. | |
|---|---|---|---|---|---|---|---|---|---|---|---|---|---|---|---|---|---|---|---|
| Simple | 6 | 108 | 25.6 | 8 | 212 | 47.2 | 9 | 18 | 13.4 | 1 | 1 | 1.0 | 6 | 18 | 8.6 | 18 | 48 | 26.5 | 17 |
| Moderate | 6 | 344 | 78.6 | 8 | 684 | 155.0 | 9 | 87 | 30.6 | 1 | 5 | 2.2 | 7 | 21 | 12.6 | 22 | 60 | 35.2 | 39 |
| Complex | 8 | 540 | 93.1 | 12 | 1092 | 184.8 | 13 | 104 | 53.9 | 1 | 7 | 3.6 | 7 | 32 | 16.9 | 25 | 82 | 47.3 | 87 |
| Very Complex | 12 | 531 | 108.0 | 20 | 1070 | 214.4 | 13 | 188 | 68.5 | 1 | 11 | 4.5 | 6 | 46 | 21.3 | 18 | 117 | 53.6 | 57 |
| All | 6 | 540 | 88.8 | 8 | 1092 | 175.7 | 9 | 188 | 50.13 | 1 | 11 | 3.4 | 6 | 46 | 16.6 | 18 | 117 | 45.0 | 200 |

## 4   *CADPrompt* DATASET

For evaluating CADCodeVerify on CAD code generation, we introduce a new benchmark the *CADPrompt*, which consists of 200 3D objects, represented both in images and STL. Each object in CADPrompt is annotated with ground truth code, and a natural language prompt.

### 4.1   PROMPT CREATION

To construct *CADPrompt* dataset, we first selected 200 3D objects from a collection of modular CAD objects from previous work  (Wu et al., 2021). Each object was manually annotated with a natural language prompt. For difficult-to-describe objects, two annotators independently provided prompts. An independent third annotator then selected the more suitable prompt. Finally, a fourth independent reviewer, not involved in the original annotation, verified and refined each of the 200 prompts to ensure accuracy and grammatical correctness. Table 1 provides the statistics of *CADPrompt*.

### 4.2   CODE ANNOTATION

We recruited a CAD Design expert to annotate CADPrompt with ground truth code for each object. The CAD design expert was provided with the language description of the 3D object, the object in STL format, and its geometric properties generated by the geometric solver (Figure 6). We then used Blender (Flavell, 2011) to validate the 3D objects produced by the expert's Python code against the ground truth and to evaluate their geometric properties (Figure 18). In cases where discrepancies were identified, the Python code was returned to the CAD expert for additional refinement. Figures 16 and 17 provide examples of Python code from *CADPrompt*.

### 4.3 DATA STRATIFICATION

We stratify *CADPrompt* examples by mesh complexity, geometric complexity and compilation difficulty to gain insights into model performance (§6).

**Mesh complexity:** We define "mesh complexity" as the total number of faces and vertices of an object. The ground truth of the 3D object is represented in mesh file formats, which consist of a collection of vertices, edges, and faces $(x, y, z)$ used to create the 3D object in the STL format. Objects with more faces and vertices are classified as more complex, while those with fewer faces and vertices are considered simple (Table 1). We split the dataset into two groups based on the median complexity: (i) Simple (those with fewer faces and vertices than the median) and (ii) Complex objects (those with more).

**Compilation difficulty:** We define "compilation difficulty" to be a measure of how difficult it is for a set of three language models (i.e., GPT-4, Gemini, and CodeLlama) to generate code for a given 3D object across two prompting methods (i.e., zero- and few-shot prompting), for a total of six attempts to generate compilable code. 3D objects were labeled then as either (i) Easy (at least four of six methods generated compilable code) and (ii) Hard (otherwise). See Appendix Figure 15.

**Geometric complexity:** We enlisted a CAD design expert to evaluate the "geometric complexity" of each object in CAD-Prompt. Each object was assigned one of the following levels: (i) Simple: the object is basic, with few features. It may consist of one geometric shape; (ii) Moderate: the object has a moderate amount of detail, with a few distinct features or components; (iii) Complex: the object has many interconnected parts, fine details, or intricate shapes; and (iv) Very Complex: the object is highly intricate, with many components, detailed textures, or complex shapes. It may have a large number of fine details, interlocking parts, or unique geometric features (Figure 3).

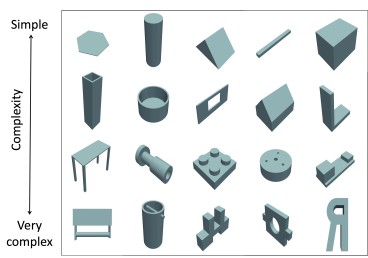

Figure 3: Examples of 3D objects from the *CADPrompt* dataset.

## 5 EXPERIMENTAL SETUP

### 5.1 BASELINES

We compare our approach, CADCodeVerify, with two baseline methods for feedback generation: (i) 3D-Premise, and (ii) Geometric solver feedback.

**3D Premise (Yuan et al., 2024):** 3D-Premise facilitates code refinement by providing GPT-4 with both the image of a generated object and its original description. GPT-4 is then prompted to correct any discrepancies between the original description and generated objects. To establish a baseline for comparison, we supply the VLM with an image of the generated object and use the same prompts as used in the original work. Note that this baseline is not applicable for approaches lacking multimodal capabilities, such as CodeLlama.

**Geometric solver feedback (this work):** We develop a novel baseline that leverages FreeCAD, an open-source geometric solver, to provide information about the geometric dimensions and structure of the generated 3D object, $d_g$. However, our proposed baseline, which computes geometric properties feedback using the geometric solver, requires access to the ground truth 3D objects, $d_{gt}$ in STL format. Specifically, the feedback, $F_{\mathcal{GS}}$, consists of numerical values across thirteen categories, $S$, each representing a unique geometric dimension or component of the object, such as width, height, number of faces, number of vertices, and volume. Formally, this feedback process is shown in Eq. 6 and 7. We compute $F_{\mathcal{GS}}$ for both the generated design, $d_g$, and ground truth, $d_{gt}$. In summary, we can describe the role of the geometric solver as follows:

$$F_{\mathcal{GS}}(d_g) = \{(s, \mathcal{GS}(s, d_g)), \forall s \in S\} \tag{6}$$

$$F_r = F_{\mathcal{GS}}(d_g) \oplus F_{\mathcal{GS}}(d_{gt}) \tag{7}$$

The feedback from the geometric solver consists of numerical values across thirteen categories of geometric information. An example of this feedback, $F_{\mathcal{GS}}$, is shown in Appendix (see Figure 6). This feedback method acts as an upper bound for CAD code refinement, as it conveys the exact geometric differences between the generated 3D object and the ground truth.

## 5.2 EVALUATION METRICS

This evaluates how accurately VLMs generate and refine 3D objects through CAD code. We use three evaluation metrics to compare the quality of the generated 3D object against the ground truth: (i) Point Cloud distance, (ii) Hausdorff distance, and (iii) Intersection over the Ground Truth (IoGT). We apply the Iterative Closest Point (ICP) algorithm to rotate and translate the generated 3D object for optimal alignment with the ground truth object (Besl & McKay, 1992). Finally, each point cloud is normalized to fit within a unit cube, as per prior work (Zheng et al., 2023). We also report the percentage of successfully compiled code as the "compile rate". The formulas for the evaluation metrics are described as follows:

**Point Cloud distance:**   Point Cloud distance $D(P,Q)$ is shown in Eq. 8, where $d_{p,q} = \|p - q\|_2$.

$$D(P,Q) = \frac{1}{2|P|} \sum_{p \in P} \min_{q \in Q} d_{p,q} + \frac{1}{2|Q|} \sum_{p \in Q} \min_{p \in P} d_{p,q} \tag{8}$$

**Hausdorff distance:**   Hausdorff distance, $H(P,Q)$, is given by Eq. 9, where $\sup$ and $\inf$ represent the supremum and infimum operators, respectively (Beauchemin et al., 1998).

$$H(P,Q) = \max\{\sup_{p \in P} \inf_{q \in Q} d_{p,q}, \sup_{q \in Q} \inf_{p \in P} d_{q,p}\} \tag{9}$$

**Intersection over the Ground Truth (IoGT):**   This metric used to evaluate how closely the generated 3D object $P$ aligns with the ground truth $Q$. As shown in Eq. 10, it calculates the ratio of the intersection area between $P$ and $Q$ to the area of $Q$.

$$IoGT = \frac{|P \cap Q|}{|Q|} \tag{10}$$

Point Cloud distance and Hausdorff distance are based on the geometric properties (e.g., height, width, and volume) of 3D objects, while the IoGT measures the overlap of the bounding boxes between the generated and ground truth 3D objects. If any generation fails to compile, we set the distance (Point Cloud and Hausdroff) to $\sqrt{3}$ (the largest possible distance between two points in a unit cube) and an IoGT value of zero (the worse possible IoGT value between two 3D objects) to maximally and uniformly penalize unsuccessful 3D object generations.

## 6 EXPERIMENT RESULTS

We utilize *CADPrompt* to quantitatively evaluate the capabilities of CADCodeVerify approach across various VLMs. In the rest of this section, we refer to the three stages of the CAD code generation process as "Generate" "Refine-1" and "Refine-2" "Generated" refers to the object generated by the LLM after the Code-Execution Step. "Refine-1" refers to the object after the first step of refinement, and Refine-2 refers to the object after the second step of refinement. Our key findings are as follows:

**GPT-4 demonstrates the highest capacity to generate compilable code *CADPrompt*.** We present our primary results in Table 2, comparing the distance and success-rate across three LLMs and three refinement methods. Regarding compile rate, GPT-4 shows superior performance at 96.5% on the *CADPrompt* benchmark, compared to Gemini at 85% and CodeLlama at 73.5%. While previous studies have provided valuable qualitative insights into GPT-4's ability to produce compilable code (Makatura et al., 2023), our analysis using *CADPrompt* offers a concrete evaluation of each LLM's CAD code generation capability.

**CADCodeVerify demonstrates superior performance on more challenging data.**   Figure 4 shows the compile rates for each feedback mechanism at the generation and refinement stages, using the difficulty and complexity splits of *CADPrompt*. For "Easy" data, the performance across all

Table 2: This table reports the median (IQR) benchmarking results for baselines across metrics. The * symbol indicates that the geometric solver accesses the ground truth to compute the geometric differences between the ground truth and the generated 3D object.

| Model | Feedback Mechanism | IoGT ↑ | Point Cloud dist. ↓ | Hausdorff dist. ↓ | Compile Rate ↑ |
|---|---|---|---|---|---|
| GPT-4: Zero-shot | Generated | 0.935 (0.043) | 0.153 (0.146) | 0.484 (0.405) | 92.0% |
| | 3D-Premise | 0.939 (0.034) | 0.150 (0.143) | **0.440 (0.372)** | 91.5% |
| | CADCodeVerify (Ours) | **0.941 (0.034)** | **0.132 (0.137)** | 0.455 (0.354) | **94.0%** |
| | Geometric solver* | **0.943 (0.037)** | **0.102 (0.159)** | 0.378 (0.434) | 91.5% |
| GPT-4: Few-shot | Generated | 0.939 (0.030) | 0.155 (0.140) | 0.494 (0.368) | 96.0% |
| | 3D-Premise | 0.942 (0.033) | 0.137 (0.155) | 0.446 (0.396) | 91.0% |
| | CADCodeVerify (Ours) | **0.944 (0.028)** | **0.127 (0.135)** | **0.419 (0.356)** | **96.5%** |
| | Geometric solver* | **0.944 (0.031)** | **0.103 (0.152)** | 0.399 (0.433) | 95.5% |
| Gemini: Zero-shot | Generated | 0.905 (0.088) | 0.159 (0.180) | 0.531 (0.451) | 85.0% |
| | 3D-Premise | 0.911 (0.079) | 0.150 (0.180) | **0.496 (0.431)** | 83.5% |
| | CADCodeVerify (Ours) | **0.914 (0.082)** | **0.138 (0.165)** | 0.497 (0.384) | **84.5%** |
| | Geometric solver* | **0.917 (0.070)** | **0.113 (0.188)** | 0.416 (0.458) | 83.5% |
| Gemini: Few-shot | Generated | 0.933 (0.061) | 0.171 (0.174) | 0.521 (0.426) | 85.0% |
| | 3D-Premise | 0.939 (0.070) | 0.169 (0.184) | 0.521 (0.516) | 81.5% |
| | CADCodeVerify (Ours) | 0.939 (0.052) | **0.147 (0.160)** | **0.492 (0.358)** | **85.0%** |
| | Geometric solver* | **0.944 (0.060)** | **0.104 (0.146)** | 0.386 (0.470) | 84.5% |
| CodeLlama: Zero-shot | Generated | 0.920 (0.943) | 0.237 (1.591) | 0.731 (1.270) | 64.5% |
| | CADCodeVerify (Ours) | **0.930 (0.955)** | **0.211 (1.599)** | **0.641 (1.309)** | **70.0%** |
| | Geometric solver* | 0.888 (0.941) | 0.280 (1.595) | 0.823 (1.258) | 55.5% |
| CodeLlama: Few-shot | Generated | 0.927 (0.949) | 0.224 (1.597) | 0.657 (1.294) | 67.0% |
| | CADCodeVerify (Ours) | **0.935 (0.957)** | **0.185 (1.620)** | **0.582 (1.366)** | **73.5%** |
| | Geometric solver* | 0.906 (0.949) | 0.239 (1.606) | 0.727 (1.301) | 60.5% |

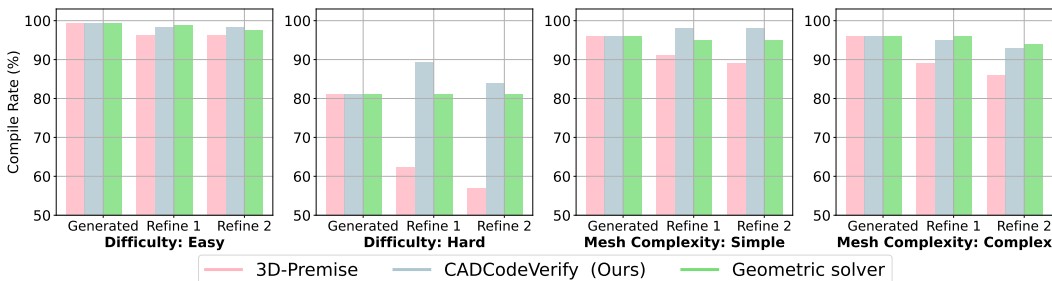

Figure 4: Compile rate comparison across different feedback mechanisms, categorized by difficulty and mesh complexity. Each figure corresponds to a specific subset of the *CADPrompt* dataset, covering both generated and refined objects. The plots for the "Hard" and "Complex" splits indicate that the feedback approach from 3D-Premise reduces GPT-4's compile rate in object generation.

baselines is relatively similar. However, results on the "Hard" data underscore the effectiveness of CADCodeVerify, which is the only refinement approach that enhances the compile rate of object generation, achieving approximately a 9% increase at the Refine-1 stage. In contrast, 3D-Premise struggles to provide effective feedback for "Hard" data, resulting in a 20% drop in compile rate at Refine-1, bringing it down to 62%. Real-world data in design and manufacturing is likely to align more closely with the "Complex" and "Hard" data splits, underscoring the value of CADCode. Our findings underscore the value of CADCodeVerify in delivering feedback that can be applied to real-world product design.

**CADCodeVerify generates model-agnostic feedback to improve 3D object generation.** Table 2 presents the distance measure from the ground truth at the generation and refinement stages. Since CodeLlama lacks multimodal capabilities, we employ GPT-4 to execute CADCodeVerify and generate refinement feedback. Our results indicate that CADCodeVerify improves the quality of the generated object across all three LLMs/VLMs as measured by IoGT, Point Cloud distance and Hausdorff distance. CADCodeVerify also increases both GPT-4 and CodeLlama's "compile rate" in producing compilable CAD code. This result highlights the model-agnostic nature of the multimodal "assess

and reason" analysis induced by our CADCodeVerify refinement process, to universally provides actionable feedback for correcting object generation errors.

Table 3: Median (IQR) results from our ablation study conducted on a randomly selected subset of 100 samples.

| Ablation Study | IoGT ↑ | PLC distances ↓ | Hausdorff dist. ↓ | Compile Rate ↑ |
|---|---|---|---|---|
| Generated | 0.909 (0.062) | 0.156 (0.150) | 0.491 (0.348) | 96.5% |
| Refine without images | 0.914 (0.058) | 0.153 (0.123) | 0.451 (0.321) | 96.0% |
| Zero-shot QA generation | **0.919 (0.049)** | 0.141 (0.112) | 0.471 (0.280) | **98.0%** |
| CADCodeVerify (Ours) | **0.919 (0.045)** | **0.126 (0.122)** | **0.444 (0.308)** | 97.5% |

**CADCodeVerify outperforms 3D-Premise across VLMs.** In Table 2, we quantitatively compare our CADCodeVerify approach to a refinement method proposed in prior work, 3D-Premise. Since 3D-Premise requires directly uploading an image of the generated object to the VLM, this approach cannot be applied to the unimodal CodeLlama model. For GPT-4 and Gemini, CADCodeVerify is shown to refine objects more accurately than 3D-Premise for objects in the *CADPrompt* dataset ($> 7\%$ reduction in point cloud distance for GPT-4 and Gemini). This finding establishes CADCodeVerify as the new state-of-the-art for CAD code refinement. Next, we compare our proposed refinement approach to an upper benchmark that uses a geometric solver to provide parametric feedback on specific geometric differences relative to the target object, as per § 5. Our results show that through CADCodeVerify, we can achieve comparable performance without requiring access to the target object, which is typically unavailable in real-world applications.

**Ablation analysis.** We conduct an ablation to evaluate the effectiveness of each independent component of CADCodeVerify (see Table 2). This analysis is performed using GPT-4 with a few-shot prompt on 100 randomly selected examples from *CADPrompt*. First, we test the effect of removing the few-shot example questions in the question-generation component (see Eq. 3). The results indicate that zero-shot QA generation increases the distance between the ground truth and refined 3D objects from 0.126 to 0.141 in Point Cloud distance and from 0.444 to 0.471 in Hausdorff distance. Our second ablation measures the impact of the reference images, $I_{ref}$, during the code refinement step of Eq. 3. Excluding the reference images worsens the Point Cloud and Hausdorff distances from 0.126 and 0.444 to 0.153 and 0.451, respectively, emphasizing the importance of the reference images.

Table 4: Performance Comparison of CADCodeVerify and Human-in-the-Loop (HITL) approaches on a subset of 50 examples from the GPT-4 few-shot setting.

| Model | Feedback Mechanism | IoGT ↑ | Point Cloud dist. ↓ | Hausdorff dist. ↓ | Compile Rate ↑ |
|---|---|---|---|---|---|
| | Generated | 0.930 (0.043) | 0.156 (0.138) | 0.495 (0.287) | 98.5% |
| GPT-4: Few-shot | CADCodeVerify | **0.948 (0.036)** | 0.137 (0.136) | 0.445 (0.302) | 98.5% |
| | Human-in-the-Loop | 0.944 (0.032) | **0.120 (0.140)** | **0.397 (0.354)** | **99.0%** |

**Gold-Standard human feedback outperforms CADCodeVerify slightly.** This experiment evaluates the performance of CADCodeVerify compared to a human-in-the-loop approach, where instead of using CADCodeVerify for code refinement, we provide gold standard language feedback which includes the exact changes that need to be made to the object. We performed this experiment on a randomly selected subset of 50 examples using the GPT-4 few-shot setting. The experiment involves two steps: (1) a human participant selects the image with the best viewing angle of the 3D object from four options (see Eq. 3), and (2) the human provides written feedback. The results indicate that the human-in-the-loop approach led to a slight improvement in Point Cloud distance and Hausdorff distance, with performance increasing from 0.137 and 0.445 to 0.120 and 0.397, respectively (see Table 4). The result of the human-in-the-loop contextualizes the performance of CADCodeVerify with respect to a method which injects domain-expertise to provide the best possible feedback. While CADCodeVerify makes significant strides in refining 3D objects, without any requiring domain expertise, some improvements are still needed to reach the level of a human expert.

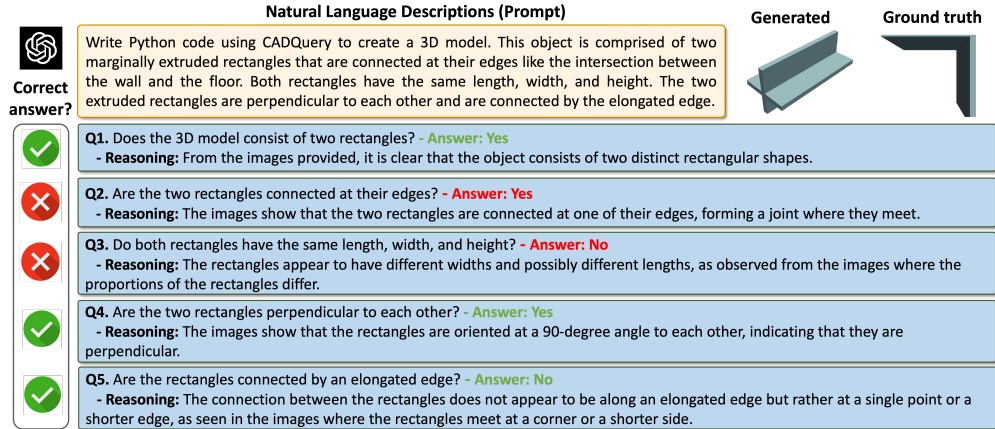

Figure 5: An example of a subset of questions and answers generated by CADCodeVerify.

**Accuracy of generated answers.** In the Question-Answering phase, CADCodeVerify is prompted to respond with "Yes," "No" or "Unclear," in situations when there is insufficient information (See Figure 5). To evaluate the accuracy of these answers, we randomly selected a subset of 50 examples from the GPT-4 few-shot setting, then manually validated the answers for both refinement stages. The results indicate that CADCodeVerify provides correct answers with an accuracy of 64.6% for Refine 1 and 68.2% for Refine 2 (see Table 9). To reduce hallucinations from the LLMs, we instructed it to respond with "Unclear" whenever it lacked confidence in its answers. In future work, we aim to explore how LLMs can interpret 3D objects and investigate methods to teach LLMs to self-verify their generated responses.

# 7    LIMITATIONS

It is essential to recognize that Point Cloud distance and Hausdorff distance are noisy metrics, measuring only the spatial similarity between two 3D objects. While they provide a broad estimate of similarity, a more granular metric is needed to capture structural differences between objects. For instance, a desk and a desk with small gaps between the legs and surface might have a low distance measure, but these gaps represent critical logical or structural issues that should be captured in the evaluation. In future work, we plan to investigate evaluation methods for CAD code generation that incorporate logical design principles.

The quality of generated objects and the likelihood of successful code compilation are influenced by the user's initial prompt. Due to the flexibility of natural language, the interpretation of 3D objects can vary, resulting in multiple valid but functionally different descriptions for the same object. For example, one could describe a desk could be described functionally as "Draw a desk with four legs," or more geometrically as "Draw an object with a flat rectangular top supported by four long rectangular prisms at each corner". In our approach, we incorporated review procedures to improve data quality, though it was not feasible to exhaustively explore all possible annotation methods. Future research could delve deeper into this prompt sensitivity and explore strategies for identifying the most effective prompts for specific VLMs.

# 8    CONCLUSION

In this work, we formally define a novel task, CAD code generation, wherein VLMs are employed to generate code for 3D parametric models. We compiled a novel *CADPrompt* dataset, comprising 200 3D objects paired with corresponding language descriptions and Python code. Next, we introduce a novel approach for code refinement within CAD code generation, called CADCodeVerify, which enables a VLM to validate and correct its generated object to address any errors in the output. We compare CADCodeVerify to two other relevant approaches for code refinement in CAD code generation, highlighting the strengths and limitations of each method. Our approach represents a substantial improvement over previous CAD code generation methods, which primarily relied on manual human feedback through language interactions with VLMs.

# 9 ACKNOWLEDGMENT

This work was supported by the National Science Foundation under grant number CMMI-2229260. We extend our gratitude to Jamie Adams, a CAD design expert, for annotating our dataset using Python code.

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

## A  ETHICS DISCUSSION

As our approach utilizes VLMs, we need to be cognizant of the potential pitfalls of utilizing these approaches. VLMs are capable of hallucinations to produce context that is unrelated or unhelpful to the desired context. These hallucinations may also include harmful propagation of the inherent

stereotypes within the datasets utilized to train these models. While these concerns are important to note for any VLM-based approach, the potential downstream impact on users via our approach is minimal. Since our approach is employed to produce 3D objects, there is limited harm which may be incurred by any potential hallucinations via our approach. However, a malicious user may choose to leverage our method to generate a 3D object of a weapon or harmful item such as guns, knives, etc. While there are many steps between the process of generating a design and procuring the item, we encourage readers to exercise caution in identifying and reporting any misuse of this approach.

## B  ADDITIONAL METHODOLOGY DETAILS

### B.1  WHY CADQUERY?

CADQuery is one of several open-source CAD scripting languages, alongside tools like FreeCAD and OpenSCAD. Previous work has primarily used OpenSCAD for CAD code generation. However, we chose to use CADQuery as our parametric CAD programming language instead of OpenSCAD, which has been the dominant language for generating 3D objects. Our decision to switch to CADQuery was based on two key reasons: (1) CADQuery is built in Python, making it more suitable for LLM-generated code, given the vast amount of Python code available online. (2) CADQuery's "design-intent" approach allows it to generate more concise code for complex objects compared to OpenSCAD.

### B.2  GEOMETRIC SOLVER VERBALIZATION

As an intermediate step, we utilize an LLM to verbalize this feedback, offering detailed insights into how the generated design $d_g$ differs from the ground truth design $d_{gt}$. Verbalization has been demonstrated to enhance the model's understanding and response accuracy (Madaan et al., 2023).

### B.3  EXPERIMENTAL PARAMETERS

We performed the experiments using GPT-4 ("gpt-v4") via the OpenAI API and Gemini ("gemini-1.5-flash-latest") through the Google API, with the temperature set to 0 for code generation and refinement. In cases where the generated code had bugs or failed to compile, we resubmitted both the code and the compiler error message to the model, adjusting the temperature to 1. For CodeLlama B70, we utilized the Replicate API[2], setting the temperature to 0.8 for code generation, refinement, and bug fixing. Other hyperparameters, such as top_k = 10, top_p = 0.9, and repeat_penalty = 1.1, were kept at their default values. The total cost for running the experiments was approximately $1000: $450 for GPT-4, $5 for Gemini, and $150 for CodeLlama. Experiments were conducted from 06 JAN to 15 FEB, 2024, and May 15 to August 15, 2024. In all our experiments, we set the number of refinements to 2, as no improvement was observed beyond the second refinement. This setting is consistent with prior work on refinement for code generation (Madaan et al., 2023; Chen et al., 2023a).

To compute distance measures, we converted the generated STL files into point clouds rendered with 1000 points. We used the Open3D and Pandas libraries to calculate the Point Cloud Distance.

### B.4  FEW-SHOT PROMPT

We design a few-shot prompt to enable VLMs to adeptly perform CAD code generation. The prompt, $p \sim y_1 \oplus y_2 \ldots \oplus y_k$, where the $y$ is CAD code and is comprised of a set of $k$ examples. We can formulate the few shot learning as $P \sim \{(y_i)\}_{i=1}^{k}$. Each example, $(y_i)$ is sourced from a memory of code samples, $D$, collected from the CADQuery documentation with a total of 40 examples. We include a text-description of the object or additional comments describing the code, in each code snippet, if it is available in the documentation.

---

[2]https://replicate.com/

## B.5 GEOMETRIC SOLVER FEEDBACK

An example of the geometric solver's feedback for both generated and ground truth 3D objects is shown in Figure 6.

```
1  {
2      "Ground_Truth": {
3          "Number_of_Faces": 140,
4          "Number_of_Unique_Edges": 210,
5          "Number_of_Vertices": 72,
6          "Width_mm": 1.5,
7          "Height_mm": 0.44,
8          "Depth_mm": 1.5,
9          "Bounding_Box_Center": "Vector (0.0, -0.22045125067234003, 0.0)",
10         "Aspect_Ratio_XY": 3.4,
11         "Aspect_Ratio_XZ": 1.0,
12         "Aspect_Ratio_YZ": 0.29,
13         "Volume_cubic_mm": 0.78,
14         "Surface_Area_square_mm": 5.59
15     },
16     "Generated": {
17         "Number_of_Faces": 500,
18         "Number_of_Unique_Edges": 750,
19         "Number_of_Vertices": 252,
20         "Width_mm": 200.0,
21         "Height_mm": 199.94,
22         "Depth_mm": 5.0,
23         "Bounding_Box_Center": "Vector (0.0, 0.0, 2.5)",
24         "Aspect_Ratio_XY": 1.0,
25         "Aspect_Ratio_XZ": 40.0,
26         "Aspect_Ratio_YZ": 39.99,
27         "Volume_cubic_mm": 157014.44,
28         "Surface_Area_square_mm": 65947.11
29     }
30 }
```

Figure 6: This figure presents an example of the feedback generated by the geometric solver, which calculates the geometric properties between the generated and ground truth 3D objects, requiring access to the ground truth.

## C  QUALITATIVE ANALYSIS

We conduct a qualitative analysis on the outputs of CADCodeVerify on a randomly selected set of 50 examples to provide further insights for two salient questions: (1) What types of feedback does CADCodeVerify generate? (2) What kinds of errors are present in the generated 3D objects?

### C.1  TYPES OF FEEDBACK GENERATED BY CADCODEVERIFY.

The feedback produced by CADCodeVerify is designed to enhance LLMs in refining the generated 3D objects. To understand the nature of the feedback generated by CADCodeVerify, we conducted a qualitative analysis and manually categorized the feedback into three main types: (i) Structural Feedback: the feedback is to correct the structure of the object (e.g., "make cylindrical or adjust corner shape); (ii) Dimensional Feedback: the feedback is an instructions related to size and scale of objects (e.g., increase height and reduce width); and (iii) Positional Feedback: the feedback focuses on the alignment of the objects (e.g., center object and align with base). illustrated in Figure 7, the percentage of Structural Feedback for generated objects starts at 52.0% in Refine 1 and decreases to 38.0% in Refine 2, demonstrating CADCodeVerify' ability to correct structural errors in the generated 3D objects. Meanwhile, Dimensional Feedback increases from 20% to 26%, likely due to some previously resolved structural errors being recategorized as dimensional issues. In future work, we aim to explore the impact of feedback types and investigate the extent to which LLMs can refine objects based on the specific type of feedback provided.

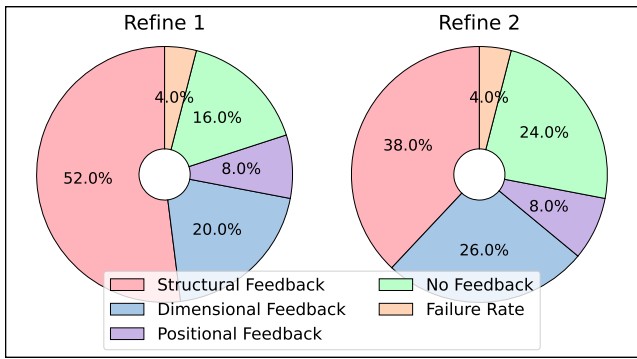

Figure 7: Analysis of the types of feedback produced by CADCodeVerify.

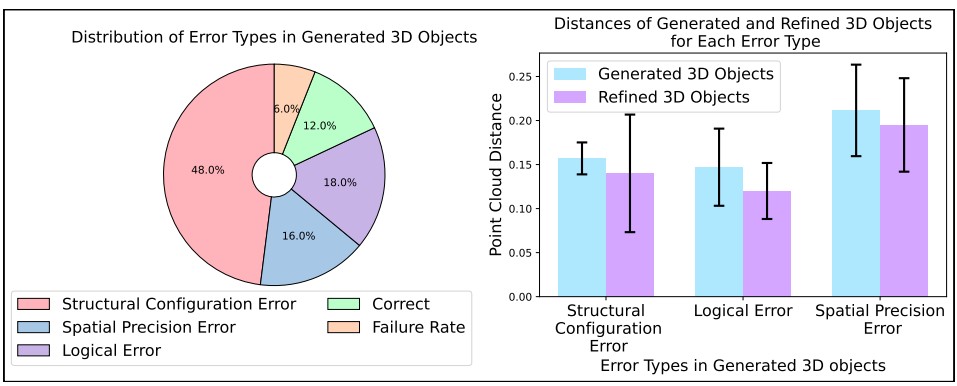

Figure 8: Analysis of errors in the generated and refined 3D objects by CADCodeVerify.

## C.2 ERRORS ANALYSIS

We conduct an in-depth analysis to categorize the types of errors present in the generated 3D objects and evaluate how much CADCodeVerify improves these objects in terms of Point Cloud distance. Following the approach in (Yuan et al., 2024), we identified five types of errors: (i) Structural Configuration Error: errors where the structure of the 3D object is incorrectly arranged; (ii) Spatial Precision Error: a minor error related to spatial parameters (e.g., height, width, and volume); (iii) Logical Error: implausible configurations of 3D objects that do not resemble real-world contexts; (iv) Correct: objects without errors; and (v) Failure Rate: objects that failed to generate due to a compile error. To identify these errors, three annotators independently categorized them, with the final annotation determined by majority vote. Figure 8 illustrates the findings: a pie chart shows that the largest proportion of errors $48\%$ are due to Structural Configuration Errors, followed by Logical Errors at $18\%$. Additionally, a bar graph compares the Point Cloud distance for generated and refined 3D objects across each error type, highlighting the improvements achieved by CADCodeVerify.

## D ALL THE RESULTS

Table 5: We present our results for GPT-4 using the median Point Cloud distance, Hausdorff distance, and Intersection over Ground Truth (IoGT), along with the interquartile range (IQR) and compile rates. Table 2 reports the Best Refine results. The * symbol indicates that the geometric solver accesses the ground truth to compute the geometric differences between the ground truth and the generated 3D object.

| Model | Feedback Mechanism | Iterations | IoGT ↑ | Point Cloud dist. ↓ | Hausdorff dist. ↓ | Compile Rate ↑ |
|---|---|---|---|---|---|---|
| GPT-4: Zero-shot | Generated | – | 0.935 (0.043) | 0.153 (0.146) | 0.484 (0.405) | 92.0% |
| | 3D-Premise | Refine_1 | 0.934 (0.035) | 0.164 (0.154) | 0.478 (0.383) | 91.5% |
| | | Refine_2 | 0.935 (0.037) | 0.170 (0.171) | 0.493 (0.436) | 89.5% |
| | | Best Refine | 0.939 (0.034) | 0.150 (0.143) | 0.440 (0.372) | 91.5% |
| | CADCodeVerify (Ours) | Refine_1 | 0.936 (0.032) | 0.146 (0.134) | 0.491 (0.362) | 94.0% |
| | | Refine_2 | 0.936 (0.039) | 0.159 (0.151) | 0.497 (0.389) | 93.5% |
| | | Best Refine | 0.941 (0.034) | 0.132 (0.137) | 0.455 (0.354) | 94.0% |
| | Geometric solver* | Refine_1 | 0.939 (0.039) | 0.119 (0.170) | 0.427 (0.470) | 91.5% |
| | | Refine_2 | 0.940 (0.047) | 0.125 (0.172) | 0.439 (0.457) | 91.0% |
| | | Best Refine | 0.943 (0.037) | 0.102 (0.159) | 0.378 (0.434) | 91.5% |
| GPT-4: Few-shot | Generated | – | 0.939 (0.030) | 0.155 (0.140) | 0.494 (0.368) | 96.0% |
| | 3D-Premise | Refine_1 | 0.937 (0.032) | 0.156 (0.176) | 0.486 (0.424) | 91.0% |
| | | Refine_2 | 0.939 (0.039) | 0.154 (0.192) | 0.467 (0.414) | 88.5% |
| | | Best Refine | 0.942 (0.033) | 0.137 (0.155) | 0.446 (0.396) | 91.0% |
| | CADCodeVerify (Ours) | Refine_1 | 0.941 (0.030) | 0.147 (0.148) | 0.470 (0.378) | 96.5% |
| | | Refine_2 | 0.941 (0.028) | 0.137 (0.139) | 0.460 (0.365) | 95.5% |
| | | Best Refine | 0.944 (0.028) | 0.127 (0.135) | 0.419 (0.356) | 96.5% |
| | Geometric solver* | Refine_1 | 0.938 (0.037) | 0.120 (0.162) | 0.429 (0.436) | 95.5% |
| | | Refine_2 | 0.941 (0.043) | 0.110 (0.162) | 0.432 (0.448) | 94.5% |
| | | Best Refine | 0.944 (0.031) | 0.103 (0.152) | 0.399 (0.433) | 95.5% |

Table 6: We present our results for Gemini using the median Point Cloud distance, Hausdorff distance, and Intersection over Ground Truth (IoGT), along with the interquartile range (IQR) and compile rates. Table 2 reports the Best Refine results. The * symbol indicates that the geometric solver accesses the ground truth to compute the geometric differences between the ground truth and the generated 3D object.

| Model | Feedback Mechanism | Iterations | IoGT ↑ | Point Cloud dist. ↓ | Hausdorff dist. ↓ | Compile Rate ↑ |
|---|---|---|---|---|---|---|
| Gemini: Zero-shot | Generated | – | 0.905 (0.088) | 0.159 (0.180) | 0.531 (0.451) | 85.0% |
| | 3D-Premise | Refine_1 | 0.903 (0.083) | 0.167 (0.192) | 0.541 (0.431) | 83.5% |
| | | Refine_2 | 0.906 (0.082) | 0.163 (0.190) | 0.548 (0.440) | 83.5% |
| | | Best Refine | 0.911 (0.079) | 0.150 (0.180) | 0.496 (0.431) | 83.5% |
| | CADCodeVerify (Ours) | Refine_1 | 0.909 (0.091) | 0.162 (0.188) | 0.527 (0.479) | 84.5% |
| | | Refine_2 | 0.906 (0.092) | 0.152 (0.170) | 0.529 (0.392) | 84.5% |
| | | Best Refine | 0.914 (0.082) | 0.138 (0.165) | 0.497 (0.384) | 84.5% |
| | Geometric solver* | Refine_1 | 0.907 (0.083) | 0.146 (0.220) | 0.479 (0.523) | 83.5% |
| | | Refine_2 | 0.910 (0.085) | 0.139 (0.203) | 0.468 (0.499) | 82.5% |
| | | Best Refine | 0.917 (0.070) | 0.113 (0.188) | 0.416 (0.458) | 83.5% |
| Gemini: Few-shot | Generated | – | 0.933 (0.061) | 0.171 (0.174) | 0.521 (0.426) | 85.0% |
| | 3D-Premise | Refine_1 | 0.934 (0.067) | 0.180 (0.193) | 0.555 (0.472) | 81.5% |
| | | Refine_2 | 0.935 (0.063) | 0.182 (0.237) | 0.576 (0.480) | 81.0% |
| | | Best Refine | 0.939 (0.070) | 0.169 (0.184) | 0.521 (0.516) | 81.5% |
| | CADCodeVerify (Ours) | Refine_1 | 0.935 (0.060) | 0.166 (0.173) | 0.541 (0.408) | 85.0% |
| | | Refine_2 | 0.932 (0.062) | 0.178 (0.187) | 0.543 (0.413) | 83.0% |
| | | Best Refine | 0.939 (0.052) | 0.147 (0.160) | 0.492 (0.358) | 85.0% |
| | Geometric solver* | Refine_1 | 0.938 (0.067) | 0.124 (0.193) | 0.433 (0.452) | 84.5% |
| | | Refine_2 | 0.937 (0.064) | 0.124 (0.208) | 0.432 (0.537) | 82.5% |
| | | Best Refine | 0.944 (0.060) | 0.104 (0.146) | 0.386 (0.470) | 84.5% |

Table 7: We present our results for CodeLLama using the median Point Cloud distance, Hausdorff distance, and Intersection over Ground Truth (IoGT), along with the interquartile range (IQR) and compile rates. Table 2 reports the Best Refine results. The * symbol indicates that the geometric solver accesses the ground truth to compute the geometric differences between the ground truth and the generated 3D object.

| Model | Feedback Mechanism | Iterations | IoGT ↑ | Point Cloud dist. ↓ | Hausdorff dist. ↓ | Compile Rate ↑ |
|---|---|---|---|---|---|---|
| Zero-shot CodeLlama | Generated | – | 0.92 (0.943) | 0.237 (1.591) | 0.731 (1.270) | 64.5% |
| | CADCodeVerify (Ours) | Refine_1 | 0.928 (0.949) | 0.223 (1.590) | 0.685 (1.281) | 70.0% |
| | | Refine_2 | 0.000 (0.939) | 1.730 (1.553) | 1.730 (1.168) | 47.0% |
| | | Best Refine | 0.930 (0.955) | 0.211 (1.599) | 0.641 (1.309) | 70.0% |
| | Geometric solver | Refine_1 | 0.888 (0.939) | 0.286 (1.590) | 0.794 (1.250) | 55.5% |
| | | Refine_2 | 0.000 (0.915) | 1.730 (1.477) | 1.730 (0.946) | 30.5% |
| | | Best Refine | 0.888 (0.941) | 0.280 (1.595) | 0.823 (1.258) | 55.5% |
| Few-shot CodeLlama | Generated | – | 0.927 (0.949) | 0.224 (1.597) | 0.657 (1.294) | 67.0% |
| | CADCodeVerify (Ours) | Refine_1 | 0.928 (0.950) | 0.212 (1.608) | 0.630 (1.324) | 73.5% |
| | | Refine_2 | 0.924 (0.946) | 0.260 (1.591) | 0.714 (1.290) | 65.0% |
| | | Best Refine | 0.935 (0.957) | 0.185 (1.620) | 0.582 (1.366) | 73.5% |
| | Geometric solver | Refine_1 | 0.903 (0.948) | 0.250 (1.604) | 0.765 (1.298) | 60.5% |
| | | Refine_2 | 0.000 (0.889) | 1.730 (1.474) | 1.730 (1.018) | 31.0% |
| | | Best Refine | 0.906 (0.949) | 0.239 (1.606) | 0.727 (1.301) | 60.5% |

Table 8: We present our results for the GPT-4 few-shot setting, stratified by object complexity (simple and complex), as discussed in §4.3. The results are reported using the median and interquartile range (IQR). The * symbol indicates that the geometric solver accesses the ground truth to compute the geometric differences between the ground truth and the generated 3D object.

| Complexity | Feedback Mechanism | IoGT ↑ | PLC distances ↓ | Hausdorff dist. ↓ | Compile Rate ↑ |
|---|---|---|---|---|---|
| Simple | Generated | 0.941 (0.017) | 0.168 (0.174) | 0.373 (0.322) | **100.0%** |
| | 3D-Premise | 0.943 (0.018) | **0.114 (0.112)** | **0.310 (0.298)** | **100.0%** |
| | CADCodeVerify (Ours) | **0.944 (0.026)** | 0.119 (0.144) | 0.329 (0.291) | **100.0%** |
| | Geometric solver* | **0.953 (0.021)** | **0.035 (0.075)** | **0.081 (0.336)** | **100.0%** |
| Moderate Complex | Generated | 0.936 (0.052) | 0.146 (0.109) | 0.465 (0.392) | 97.4% |
| | 3D-Premise | 0.942 (0.047) | **0.102 (0.107)** | **0.341 (0.331)** | **100.0%** |
| | CADCodeVerify (Ours) | **0.943 (0.035)** | 0.114 (0.124) | 0.416 (0.332) | 97.4% |
| | Geometric solver* | 0.937 (0.049) | **0.085 (0.082)** | **0.311 (0.319)** | 97.4% |
| Complex | Generated | 0.938 (0.024) | 0.157 (0.184) | 0.504 (0.387) | 94.3% |
| | 3D-Premise | 0.939 (0.029) | 0.154 (0.228) | 0.469 (0.427) | 87.4% |
| | CADCodeVerify (Ours) | **0.942 (0.022)** | **0.114 (0.133)** | **0.412 (0.297)** | **96.6%** |
| | Geometric solver* | **0.946 (0.027)** | **0.110 (0.159)** | 0.429 (0.447) | 94.3% |
| Very Complex | Generated | 0.941 (0.038) | 0.165 (0.140) | 0.541 (0.273) | 96.5% |
| | 3D-Premise | 0.946 (0.030) | 0.156 (0.198) | 0.498 (0.478) | 87.7% |
| | CADCodeVerify (Ours) | **0.954 (0.039)** | **0.144 (0.123)** | **0.475 (0.359)** | **94.7%** |
| | Geometric solver* | 0.943 (0.047) | **0.114 (0.152)** | **0.465 (0.330)** | **94.7%** |

Table 9: Accuracy of answers generated by CADCodeVerify on a subset of 50 examples in the GPT-4 few-shot setting.

| Answers | Refine 1 | Refine 2 | Description |
|---|---|---|---|
| Total answers | 219 | 176 | Total number of responses evaluated |
| Correct answers | 64.6% | 68.2% | We evaluate each answer as either "Yes" or "No" and calculate the percentage of these responses, including incorrect and "Unclear" answers. For Refine 1: Out of 214 answers labeled as "Yes" or "No," only 19 were incorrect, resulting in an accuracy rate of 91%. For Refine 2: Out of 176 answers labeled as "Yes" or "No," 22 were incorrect, yielding an accuracy rate of 87.5%. |
| Incorrect answers | 8.8% | 12.5% | The percentage of incorrect answers where the CADCodeVerify response was "Yes" or "No," but the answer was incorrect. |
| Answers with "Unclear" | 26.6% | 19.3% | To minimize hallucinations from the LLM, we instructed it to respond with "Unclear" when it lacked confidence in its answer. Any response labeled as "Unclear" was sent back to the LLM during the refinement phase for further evaluation. |

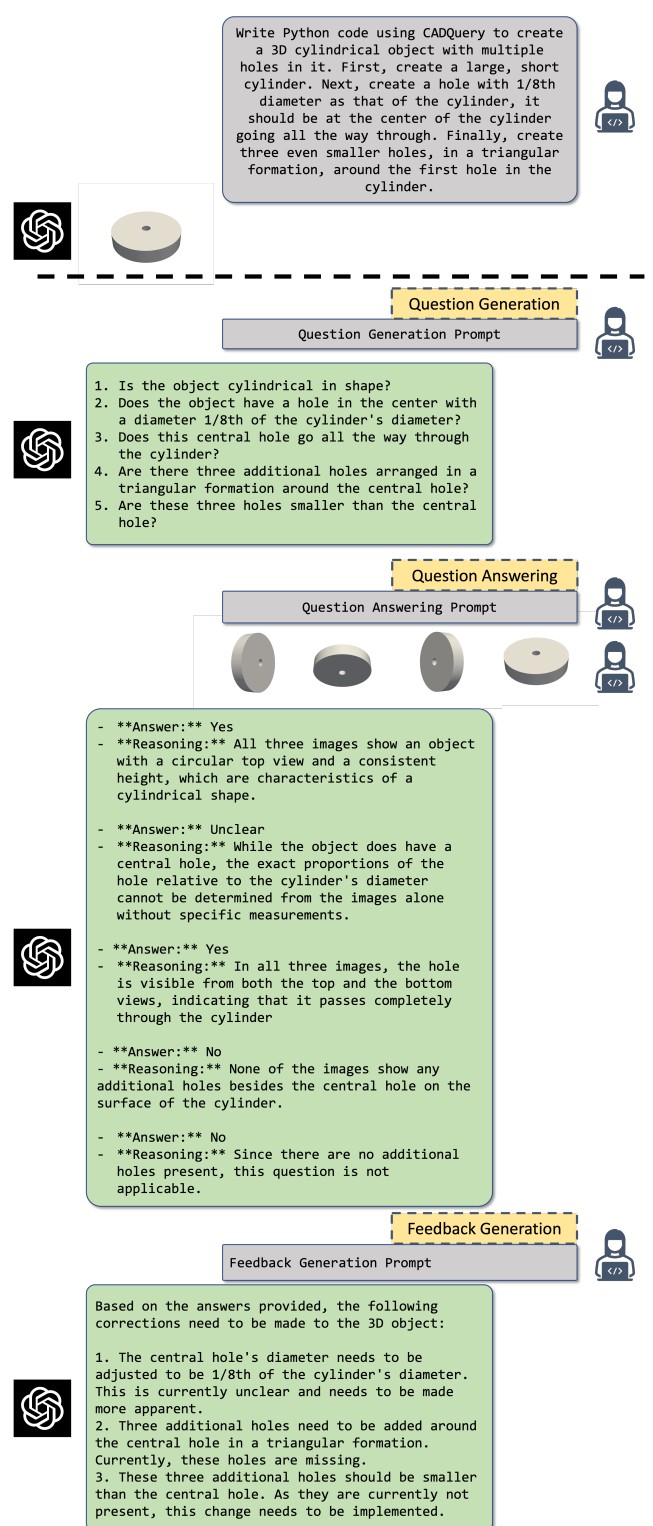

Figure 9: This figure provides an example (1) of one step of CADCodeVerify, via GPT-4V, for a prompt from *CADPrompt*

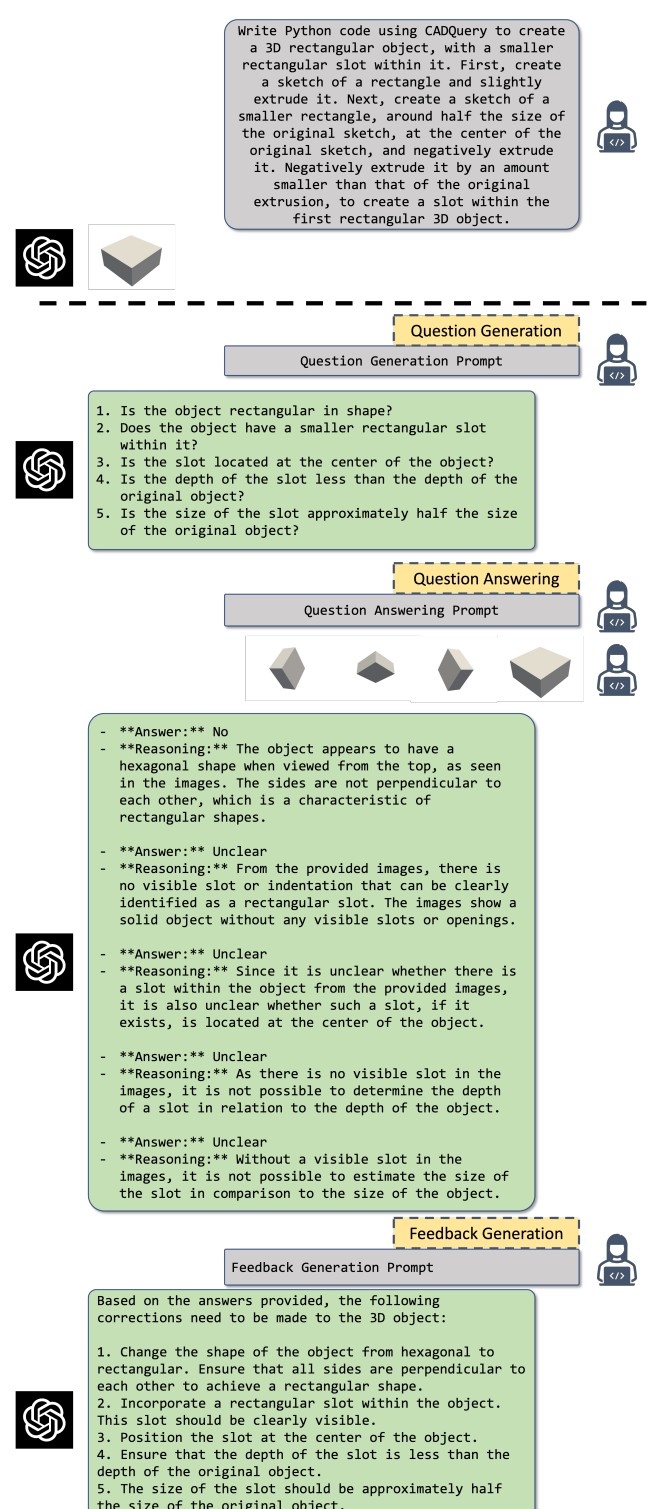

Figure 10: This figure provides an example (2) of one step of CADCodeVerify, via GPT-4V, for a prompt from *CADPrompt*

Prompt for generating questions.

You will be given a description of how a human-designer would describe the design of a 3D object. Your job is to provide between 2-5 (Yes or No) questions that I can use to verify how similar the generated object is to the description generated by a human. The questions should be framed such that answering "No" implies that there is a change that needs to be made to the object regarding the verification question. Here are some important points to note for this task;

(1) Do not make up questions if you cannot generate 5 questions based on the description provided.

(2) Ensure that your questions only reference entities mentioned within the description.

(3) Try not to reference orientation the components of the 3D object. Your generated questions should not ask whether a component is on the "right" or "left" side as this orientation is relative.

I will give you two examples with a language description followed by the appropriate verification questions. Please reference these examples while generating your verification questions.

### Example 1 ###
Description:
Extrude a cylindrical plate with a rectangular hole in the middle of it.

Generated Questions:
1. Is the object cylindrical in shape?
2. Does the object have a rectangular hole in the center?
3. Is the object extruded in one dimension?

### Example 2 ###
Description:
Design a 3D object that resembles a cone. First draw a sketch of a square and extrude it to create the base of the cone. Next, draw a sketch of a circle centered at the center of the square base. Extrude this sketch vertically into a conical shape, such that the diameter of the circle decreases as the height increases. Finally cutout the tip of the cone, such that the tip of the cone is now rectangular in shape.

Generated Questions:
1- Does the object resemble a cone?
2- Is the base of the object square-shaped?
3- Is the circular base of the cone centered at the same point as the center of the square base?
4- Is the tip of the cone rectangular?
5- Does the diameter of the cone decrease as the height increases?

Figure 11: In the CADCodeVerify approach, we utilize this prompt with two examples to generate the verification questions.

---

Prompt for code repair.

You will be provided with a piece of Python code and a compiler error message, and then
your task will be to fix the bugs and rewrite the code.

#### Compiler error messages:
In line 20:
.vertices().fillet(cutout_radius))
raise ValueError(
ValueError: Cannot find a solid on the stack or in the parent chain

#### Python Code:
```python
import cadquery as cq

# Dimensions
plate_length = 100
plate_width = 50
plate_thickness = 3
cutout_size = 15
cutout_radius = 5

# Create the base plate
plate = (cq.Workplane("XY")
.rect(plate_length, plate_width)
.extrude(plate_thickness))

# Create the cutout sketch with rounded corners
cutout_sketch = (cq.Workplane("XY")
.moveTo(cutout_size/2, cutout_size/2)
.rect(cutout_size, cutout_size, forConstruction=True)
.vertices().fillet(cutout_radius))
# Cutouts at two corners cutout1 = (plate.workplane()
.placeSketch(cutout_sketch)
.extrude(-plate_thickness))

cutout2 = (plate.workplane()
.workplane(offset=plate_length - cutout_size)
.placeSketch(cutout_sketch)
.extrude(-plate_thickness))

# Combine the base plate and cutouts
final_object = plate.cut(cutout1).cut(cutout2)

# Export the result to an STL file
# This line caused the error. ".val()" is removed
final_object.exportStl("Generated.stl")
```

---

Figure 12: If the generated CAD code fails to compile due to a compiler error, we pass both the error
and the generated CAD code to LLMs for correction.

Prompt for generating answer.

Your job is to answer this set of questions with respect to the object I have shared with you. I will be providing 4 images of the object from different orientations so that you can get a complete picture of the 3D object. Here are some important points to note regarding your task:
(1) Remember that these images are all of the same object from different angles.
(2) The answer to each of these questions should always be one of three options which are "Yes" or "No" or "Unclear."
(3) Your answer should be "Unclear" in situations where you are unsure of the answer or do not have enough information to answer the question.
Make sure to provide reasoning supporting all your answers.

# Your answer should follow the same format as below:
1. **Question?**
- **Answer:**
- **Reasoning:**

2. **Question?**
- **Answer:**
- **Reasoning:**

Figure 13: In the CADCodeVerify approach, we use this prompt along with images to generate the answer and reasoning for each question.

Prompt for generating feedback.

These were the answers to the questions I asked to validate a generated 3D object. Can you utilize the answers to these questions to generate actionable feedback that will help the model to correct the mistakes in the 3D object? Your job is to summarize these answers into practical corrections that need to be made to the 3D object. Please note the following while generating your feedback:
(1) The corrections should be such that the answers to all questions provided will become yes upon applying the suggested corrections.
(2) Your corrections should not change the object such that any of the answers that are already, "Yes" become "No."
(3) You only want to change the object such that the answers which are "No" or "Unclear" become "Yes." The summary should be specific and only a few sentences long.
(4) Your corrections should not be regarding the quality or orientation of the images.
(5) Your feedback should not attempt to fix issues in the scale. DO NOT ask for the addition of additional scale or reference objects.
(6) Do not ask for details regarding the size or dimensions of the object.
(7) Your corrections should be constructed such that a human designer can use your feedback to update the 3D object such that all questions have "Yes" as the answer.

Figure 14: In the CADCodeVerify approach, we use this prompt to generate feedback from the generated question-answer set.

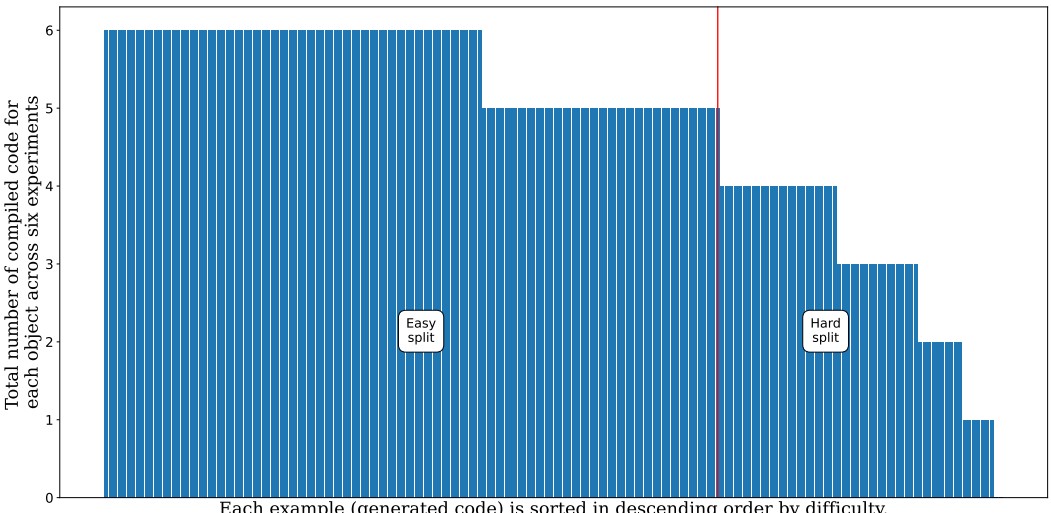

Figure 15: We conducted six experiments across three LLMs—GPT-4, Gemini, and CodeLlama—and in both zero-shot and few-shot settings for *CADPrompt*, which contains 200 examples. Some of these experiments did not generate compiled code that produced valid 3D objects for some of the examples. We calculated the total compiled code for each example across all experiments, where Max = 6 means the example was generated in all experiments, and Min = 0 indicates that the example was not generated by any experiment. We then sorted them based on difficulty and split the dataset into "Easy" and "Hard" categories as discussed in §4.3

Write Python code using CADQuery to create a triangular 3D object. First, draw a sketch of an equilateral triangle, pointing downwards. Next, cutout a semicircle from the bottom corner of the triangle. The diameter of this semicircular cutout should be approximately 2/3rd of the length of each side of the triangle. Finally, extrude this sketch to create a 3D object.

(a) The natural language descriptions of the 3D object.

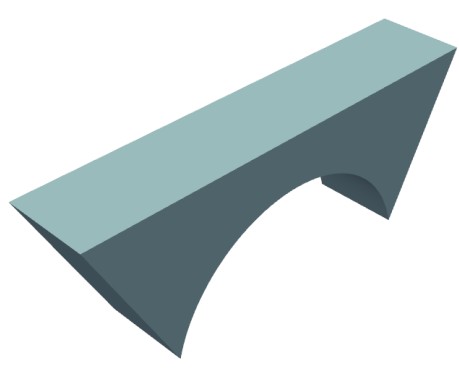

(b) 3D object

```python
import cadquery as cq
from typing import List,Tuple

length:float = 0.929516
width:float = 0.54
height:float = 0.3
top_length:float = 1.5

top_x:float = (length - top_length)/2

points:List[Tuple[float, float]] = [
    (0, 0),
    (length, 0),
    (top_x + top_length, width),
    (top_x, width),
]

work:cq.Workplane = (
    cq.Workplane("XZ")
    .center(-length/2,-width/2)
    .polyline(points).close()
)

part:cq.Workplane = work.extrude(height).translate((0,height/2,width
    /2+0.12))

radius = (length/2)+0.014

ellipse = cq.Workplane("XZ").circle(radius).extrude(height).translate
    ((0,height/2,))
part = part.cut(ellipse.translate((0,0,width+0.12-radius-0.18)))

cq.exporters.export(part, 'Code_Ground_Truth.stl')
```

(c) Python Code

Figure 16: An example from the *CADPrompt* dataset, showing (a) the prompt, (b) the corresponding 3D object, and (c) the human-annotated Python code used to generate the 3D object.

Write Python code using CADQuery to create an inverted desk, with hexagonal legs. First, draw a rectangular sketch, and cutout four small, right-angle triangles from each corner of the rectangle to create an octagonal surface. Next, extrude this sketch by a small amount. Draw two large, regular hexagons, placed symmetrically on opposite ends of the octagonal surface. The hexagons should be the same size and should align perfectly with each end of the surface. Finally, extrude these hexagons outwards to give the appearance of two hexagonal columns protruding from a horizontal surface.

(a) The natural language descriptions of the 3D object.

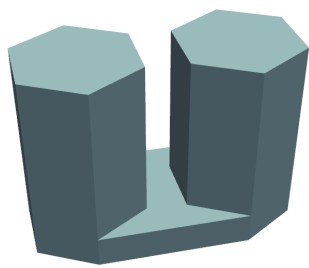

(b) 3D object

```python
import cadquery as cq

diameter = 0.5
width = 0.433013
spacer = 0.155128
sides = 6
height = 0.75

hexes = (
    cq.Sketch()
    .push([(0,0),(width+spacer,-0.018963)])
    .regularPolygon(diameter/2,sides)
)

hex_base = (
    cq.Sketch()
    .push([(0,0),(width+spacer,-0.018963)])
    .regularPolygon(diameter/2,sides)
    .wires()
    .hull()
)

hexagans = (
    cq.Workplane("XY")
    .placeSketch(hexes)
    .extrude(height)
)

base = (
    cq.Workplane("XY")
    .placeSketch(hex_base)
    .extrude(0.125)
)

part = (
    cq.Workplane("XY")
    .union(hexagans)
    .union(base)
)

part = part.rotate((0,0,1),(0,0,0),30+180).translate
    ((00.339552,-0.348832,0))

cq.exporters.export(part, 'Ground_Truth.stl')
```

(c) Python Code

Figure 17: An example from the *CADPrompt* dataset, showing (a) the prompt, (b) the corresponding 3D object, and (c) the human-annotated Python code used to generate the 3D object.

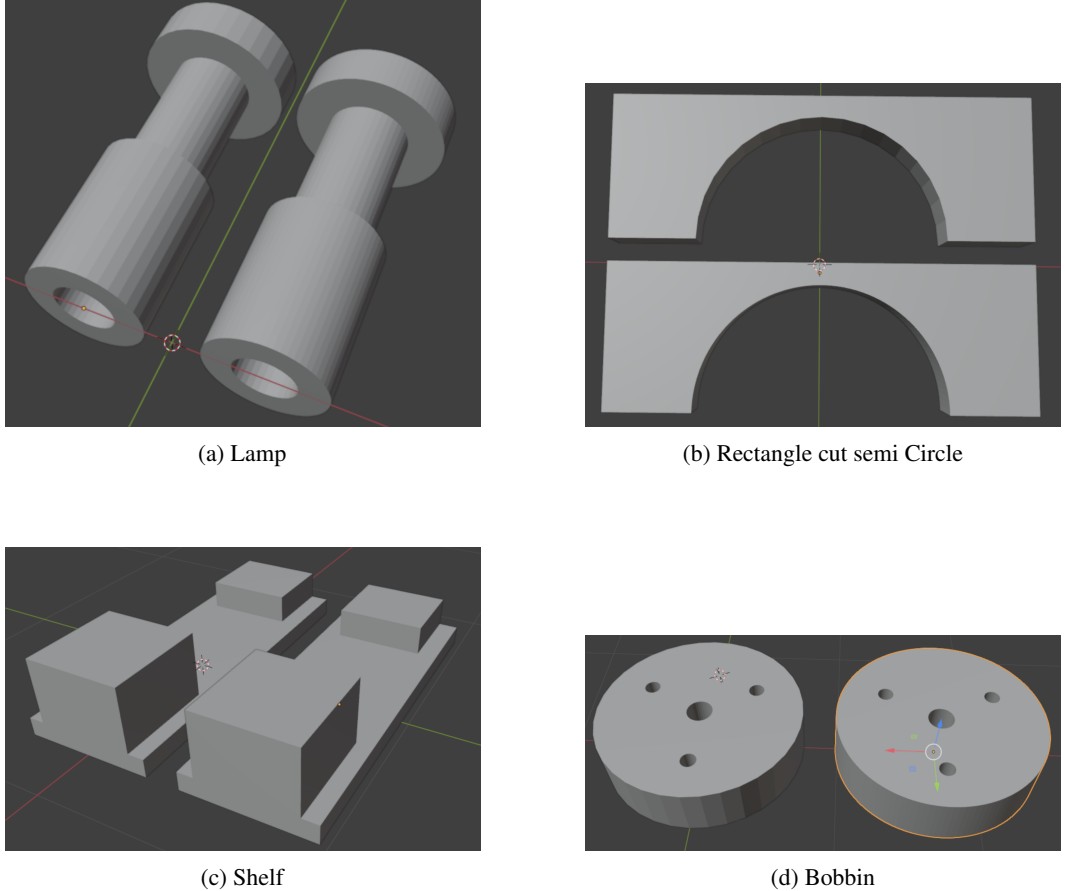

(a) Lamp

(b) Rectangle cut semi Circle

(c) Shelf

(d) Bobbin

Figure 18: Comparison of the ground truth 3D object with the 3D object generated by Python code written by a human CAD design expert.

