# OpenReview forum: "Generating CAD Code with Vision-Language Models for 3D Designs"
_ICLR.cc/2025/Conference — ICLR 2025 Poster_

### Official Review · Reviewer_9oaM · 2024-11-03

**Soundness:** 4
**Presentation:** 4
**Contribution:** 3
**Rating:** 6
**Confidence:** 4

**Summary:**

This paper proposes a technique called CADCodeVerify that uses feedback generated from VLMs to refine CAD code generated by LLMs in an iterative manner. This is a novel technique proposed to address a reasonably difficult and seemingly important problem. The paper evaluates this technique against an existing baseline called 3D-Premise and also with a baseline using feedback from a geometric solver. CADCodeVerify shows better performance against both baselines.

**Strengths:**

This paper is written really well. As someone who doesn't work much with CAD code, it was immediately clear what the limitations were and how they intended to address them. The technique itself is novel for the application and is well described with appropriate examples. I also appreciate the effort taken for creating the benchmark tasks over which the systems were evaluated.

**Weaknesses:**

One main weakness I can see is the lack of baselines. However, this is not a major issue to me since I don't know of other systems that target this specific problem. However, I can think of other techniques like chain of thought prompting, or techniques using some form of reasoning, against which we can compare CADCodeVerify. Right now, the only true baseline seems to be 3D-Premise, and another baseline that uses an alternate form of feedback generation, though I could be mistaken here.

**Questions:**

1. In Fig 1, within the CADCodeVerify block, I see step 3.c generating feedback which then produces a new object. My question is: what exactly is the input to the VLLM? Is it the feedback, that then generates the code that compiles to produce the new object, or is it the new object itself? I suspect it is the former, but the figure indicates it to be the latter.
2. In the evaluation, do you consider naive or basic prompting? For example, just a zero-shot or few-shot prompt to an LLM that produces code? Is that what the "Generated" field in Table 2 is?
3. Are there any insights as to why CADCodeVerify performs better than baselines? For instance, what is it in the proposed technique that allows it to outperform 3D-premise? And why is VLLM feedback more useful than geometric feedback, even when the latter needs the presence of ground truths? One would imagine a technique needing GT labels for providing feedback would be more accurate or equivalent to a technique operating without the labels.
4. Maybe I missed this in the text, but in Fig 3, what is Refine 1 and Refine 2?
5. Are there any limitations as to the complexity and difficulty of the benchmark tasks? I understand they vary in complexity and difficulty, but at the same time CADCodeVerify does quite well, even reaching around 90% accuracy in the hardest case. Is there a class of problems this benchmark is not testing, or are the hard problems simply not hard enough?
6. How representative is the benchmark of the objects actually being drawn in the industry today? One major challenge with code generation has always been that it has never been representative of IRL code. Do we see the same challenges here?
7. What dataset was 3D-Premise originally evaluated on, and why have you not evaluated CADCodeVerify over that? What is different about your dataset that sets it apart from the one 3D-Premise used?

---

> ### Author Response · Authors · 2024-11-21
> **Response to Reviewer 9oaM Part (1/2)**
>
> We sincerely thank the reviewers for their thoughtful feedback and valuable insights, which have greatly improved the quality of our work.
>
> ## What is the input to the VLLM during refinement
>
> Our feedback generation process is comprised of three steps (Figure 1). (a) The LLM is prompted to generate questions based on the prompt to verify whether the generated object corresponds to the prompt. (b) The LLM then analyzes the generated object and answers the set of validation questions from step-a. (c) The answers to these questions are then summarized concisely as feedback to refine the initially generated object. Therefore, to answer the reviewer’s question, the generated object is passed back into the model to generate language-based feedback. This language-based feedback is then provided to the LLM to update the initial code.
>
> ## Do we consider naive or basic prompting
>
> Yes, the results in the “Generated” phase are the results with just a basic prompt. We experiment with a zero-shot and a few-shot prompt as shown in table-2.
>
> ## Main weakness is a lack of baselines
>
> To the best of our knowledge, the only pre-existing method for CAD Code Refinement is 3D-Premise. By showing that we outperform 3D-Premise across all four metrics, we show that CADCodeVerify is the new state-of-the-art. We further develop a second baseline which compares our method which provides structural feedback, to a method which provides precise feedback (Geometric Solver) regarding the specific parametric differences between the two objects. Our methodology also integrates in-context learning methods such as Chain-of-thought reasoning, few-shot exemplars, and self-verification. If the reviewer has specific baseline suggestions in mind, we are willing to implement and include them in the paper. However, we believe our work makes a significant technical contribution and is thoroughly evaluated against existing methods.
>
> ## Why CADCodeVerify performs better than 3D-Premise
>
> 3D-Premise simply prompts the model to utilize an image of the generated object and update the code based on any discrepancies identified. Whereas CADCodeVerify first prompts the model to "self-verify" and produce actionable feedback, which is provided to the VLM in conjunction with the image of the object to better isolate the changes that need to be made to the object. Recent work on visual programming has also shown that when utilizing visual feedback to update or refine code, it is helpful to process the images and produce textual-information about the image to integrate into the refinement feedback [1].
>
> [1] - Gao, Minghe, et al. "De-fine: De composing and Re fin ing Visual Programs with Auto-Feedback." Proceedings of the 32nd ACM International Conference on Multimedia. 2024.
>
>
>
> ## Why CADCodeVerify performs better than the Geometric Solver
>
> We also found this result to be interesting! The Geometric solver provides high-level geometric feedback about the object (See Figure 6 in the appendix), including details about the volume, surface-area, height, width, etc. However, the feedback from the geometric solver does not provide any insights regarding the structural correctness of the object. CADCodeVerify on the other hand explicitly focuses on correcting structural errors in the object relative to the prompt provided. The first example shown in figure 2 provides a depiction of this. The generated prism has very similar geometric properties with regards to the categories expressed by the geometric solver, therefore it is unable to adequately discern and correct the errors in the generation. CADCodeVerify on the other hand identifies that the square cutout should be moved “to the top-edge of the rectangle” as stated in the original prompt and is able to produce feedback to address this discrepancy.
>
> ## Refine 1 / 2 definitions
>
> “Refine-1” and “Refine-2” correspond to the results after the first and second refinement steps.

---

> > ### Author Response · Authors · 2024-11-21
> > **Response to Reviewer 9oaM Part (2/2)**
> >
> > ## How representative are the objects in your dataset?
> >
> > We have conducted an additional experiment to understand the breakdown of the semantic complexity of the objects in our dataset. We recruited an independent annotator (who is a mechanical engineer) to annotate the structural complexity of each of the 200 objects in CADPrompt according to the following scale:
> >
> > - (Simple) - The object is extremely basic, with few features. It may
> > consist of one geometric shape.
> >
> > - (Moderately Complex) - The object has a moderate amount of detail,
> > with a few distinct features or components
> >
> > - (Complex) - The object is complex, with many interconnected parts,
> > fine details, or intricate shapes.
> >
> > - (Very Complex) - The object is highly intricate, with many
> > components, detailed textures, or complex shapes. It may have a large
> > number of fine details, interlocking parts, or unique geometric features
> >
> > The breakdown of our dataset according to this independent reviewer’s evaluation was as follows: Simple - 17, Moderate Complexity - 39, Complex - 87, and Very Complex - 57. Please refer to Table 1 in the updated version.
> >
> >
> >
> > In terms of real-world representativeness, real-world manufacturable designs generally fall under the “very complex” category and upwards. Currently we observe that existing LLMs can easily generate simple/moderate objects through existing in-context learning methods. However, consistently generating very complex objects at high-degrees of precision is still an unsolved problem. CADPrompt is a test-bed which provides a stepping-stone to test the competency of CAD Code Generation methods prior to deployment on real-world objects.
> >
> > ## Why didn’t we evaluate on the dataset used by 3D-Premise
> >
> > The dataset provided by 3D-Premise was not made publicly available by the authors, which prompted us to create CADPrompt. Furthermore, 3D-Premise's dataset was only comprised of 57 examples, with insufficient information regarding how the annotations and objects were collected and validated, which impeded our ability to reproduce a similar dataset. Leveraging LLMs for CAD model generation is still a nascent area of research, with the first approach introduced as recently as May 2023. As such, there is a dearth of standardized testing methods and benchmark to competently evaluate the capabilities of CAD Code Generation methodologies. Through CADPrompt, we provide a much needed, publicly available benchmark to standardize future work on CAD Code Generation.

---

> ### Comment · Reviewer_9oaM · 2024-11-25
>
> Thanks for the clarifications. I will keep my score

---

> > ### Author Response · Authors · 2024-11-25
> > **Response to Reviewer 9oaM**
> >
> > Dear 9oaM,
> >
> > Thank you for your valuable feedback and response. As we approach the end of the discussion period today, we welcome any final thoughts or concerns about our paper that you would like us to address. Please don’t hesitate to share any additional feedback or questions—we are happy to discuss further and work toward improving your evaluation of our paper.

---

### Official Review · Reviewer_8EJ1 · 2024-11-04

**Soundness:** 2
**Presentation:** 3
**Contribution:** 2
**Rating:** 6
**Confidence:** 3

**Summary:**

This work contributes a new method for VLM-based CAD code synthesis, and a new dataset for evaluating CAD programming models. It builds on past work on code repair for CAD synthesis – the core idea is that instead of just providing the current render during repair (as prior work 3D-Premise did), the LLM also comes up with *verification questions* based on the task description (not using the current render) which it then answers (using the current render). These Q/A pairs are summarized into feedback for what needs changing, which is provided to the repair model alongside the current render to prompt code repair.

The dataset, CADPrompt, was constructed by selecting objects from a dataset used in prior work and annotating them with language prompts along with Python CAD code. They evaluate on a range of models (GPT4, Gemini, CodeLLama) and compare to several baselines (no refinement, 3D-Premise, and Geometric Solver). The last of these baselines, Geometric Solver, is also a contribution – it's a baseline that gets to "cheat" and compare the ground truth model to the current rendered model in terms of a range of geometric feaures. Their methods generally outperforms the baselines.

**Strengths:**

- The contribution of a language-to-CAD dataset is useful for the community, and hiring CAD experts to write the CAD code for it means it's likely high quality.
- The comparison is run across a range of LLMs including open source ones (CodeLlama)
- They generally surpass the 3D-Premise baseline and score fairly close to the geometric solver baseline that gets to cheat and use the ground truth CAD model in calculating its feedback. They score particularly well on the more difficult problems, and benefit more from feedback than 3D-Premise. They show improvement across three different similarity metrics in addition to compilation success rate.

**Weaknesses:**

- The term "success rate" is a bit of a misleading name for something that means "compilation rate" or "compilation success rate" – it gives me the impression that the model succeeded at solving the task, not that it generated an arbitrary piece of code that compiled. For example in the abstract "increasing the success rate of the compiled program" or in the intro "5.5% increase in successful object generation" all gave me this impression until I dug into it.
    - Also, the abstract says 5.0% and the intro says 5.5% – which is correct?
- Does 3D-Premise get to do the Code Execution step described in 3.2, where if compilation fails it retries with compiler feedback up to N times? If not, it seems important to do this comparison to disentangle how much of the benefit is coming from compiler feedback versus the Q/A contribution.
- The improvements are modest but still reasonable – I would defer to a reviewer more familiar with these metrics on this. Looking at GPT-4 Few-shot in Table 2 (which looks like overall the best model), the IoGT metric difference is .942 vs .944, the Point Cloud distance is .127 vs .137, and the Hausdorff distance is .446 vs .419. These are not huge differences, but still a notable step towards the Geometric Solver performance.
- It's confusing that "Refine 1" causes 3D-Premise to decrease in performance in Figure 3 – see full note in "Questions" section, but clarification on this would be helpful to understanding the results.

**Questions:**

- Is the Geometric Solver output given to an LLM during the feedback phase? I assume so given how it varies depending on which LLM is used in Table 2. Does it get to do two rounds of feedback like the CADCodeVerify model does?
- "Refine 1", "Refine 2", and "Generated" should be defined somewhere, right now the first place they appear is in Figure 2 headings (which are not searchable text in the PDF) and then when discussing results, but they aren't clearly defined.
    - Is "Generated" the step right after Section 3.1 finishes or right after Section 3.2 finishes – i.e. is "Generated" before or after the compiler-error based verification step?
    - Are "Refine-1" and "Refine-2" the two rounds of refinement (Section 3.3) or is the first one the code repair from Section 3.2?
- Why does "Refine-1" go *down* so much in Figure 3 for 3D-Premise? This is very surprising to me and would be helpful to clarify. Refinement ought to help, and it did help them in that prior work. While it's not as strong of an effect, refinement also seems to hurt all the methods a bit in some of the other plots (like the Complex plot) – why would rounds of refinement hurt instead of help?
    - These graphs are showing the compilation rate. It'd be helpful to see changes in the other metrics from Table 2 that have to do with correctness instead of whether or not the code compiles – those seem key to understanding whether feedback is helping.

Minor fixes:
- The CADQuery compiler $\phi$ is deterministic, right? So the "$\sim$" should be an "=" on line 177
- The description of the "number of compilable outputs" discussed in 305-310 was confusing to me until I saw the caption of Figure 11 in the appendix. It would be helpful to move part of this description to the main paper – it wasn't clear to me that this meant the number of experiments where the code compiled successfully. And to clarify – are those 6 experiments the six large rows in Table 2 (that vary between language model used and not between method used)?
- In discussing the geometric solver the paper notes "This feedback method serves as an upper limit for CAD code refinement, as it conveys the precise geometric differences between the generated 3D object and the ground truth." While I think this is a useful baseline, it's not literally an upper limit for performance from code refinement feedback – as shown for example by how it's occasionally outperformed in the experiments. It's a great baseline to have, I just would suggest against using the term "upper limit" for something that's not actually an upper bound.

---

> ### Author Response · Authors · 2024-11-21
> **Response to Reviewer 8EJ1**
>
> We thank this reviewer for their time and valuable feedback to our submission.
>
> ## “Success-Rate” is misleading
> We thank the reviewer for highlighting the potential misinterpretation of the term “success-rate.” We will change “success-rate” to “compile-rate” as per the reviewer’s suggestion.
>
> ## 3D-Premise execution step
> Yes, 3D-Premise goes through the code-execution step in the same manner as CADCodeVerify and the Geometric Solver baseline. The baselines only differ in approach to ours in the Code-Refinement Step (Section 3.3). Both 3D-Premise and the Geometric solver approach also go through N-steps of code-repair to engender a consistent comparison with CADCodeVerify.
>
> ## Refine-1 causes 3D-Premise to worsen compile-rate
>
> We hypothesize that 3D-Premise leads to a reduction in the compile-rate, because off-the-shelf LLMs struggle to infer the changes they need to make to CAD scripting code to edit the 3D object, based on the image in isolation. In contrast, CADCodeVerify includes both generated images of the object as well as the associated text-feedback, computed via self-verification. This text-feedback likely provides “instructions” to enable the VLM to better interpret the changes it needs to make to the code based on the image, and better aligns with the types of feedback the model has been trained on. Recent work on visual programming adopts a similar approach wherein they extract textual interpretations from visual features rather than utilizing the images in isolation [1]
>
> [1] - Gao, Minghe, et al. "De-fine: De composing and Re fin ing Visual Programs with Auto-Feedback." Proceedings of the 32nd ACM International Conference on Multimedia. 2024.
>
> ## Geometric Solver Feedback
>
> Yes, the output from the geometric solver is passed back into the model as feedback to refine the initial generated code. We verbalize the raw outputs from the Geometric solver, prior to passing it into the LLM as feedback (Section B.2 in the appendix). Both the Geometric solver method and 3D-Premise utilize two feedback steps, similar to our approach.
>
> ## Generated, Refine-1, Refine-2 definitions
>
> Please pardon any confusion regarding the terminology used in the experiments section. We have added these defintions to the first paragraph of Section 6  --  “Generated refers to the object generated by the LLM after the Code-Execution Step. Refine-1 refers to the object after the first step of refinement, and Refine-2 refers to the object after the second step of refinement. “
>
> ## Additional Questions/Typos
>
> - The number in the introduction is correct (5.5%). We will update the abstract to correct this typo
> - Mistake in the equation on line 177 has been corrected.
> - We will remove “upper-limit/upper-bound” while referencing the Geometric solver baseline.
> - The three experiments discussed between lines 305-310 reference the 2x3 configuration of code generation approaches across the axes of model-type (GPT-4, Gemini, CodeLlama) and prompt-type (Zero-shot/Few-shot).

---

> > ### Author Response · Authors · 2024-11-25
> > **Response to Reviewer 8EJ1**
> >
> > Dear Reviewer 8EJ1,
> >
> > We hope you’ve had an opportunity to review our responses to your valuable feedback. As the discussion period concludes today, we wanted to follow up to see if there are any remaining concerns or points of clarification that you would like us to address. Please don’t hesitate to share any additional feedback or questions—we are happy to discuss further and work towards improving your evaluation of our work.
> >
> > Thank you again for your time and effort in reviewing our paper.

---

> > > ### Comment · Reviewer_8EJ1 · 2024-11-25
> > >
> > > > We hypothesize that 3D-Premise leads to a reduction in the compile-rate, because off-the-shelf LLMs struggle to infer the changes they need to make to CAD scripting code to edit the 3D object, based on the image in isolation.
> > >
> > > Ah, and I see now how in Table 2 the compile rate for 3D-Premise is lower than that of Generated, which is showing this same trend that appears in Figure 4. And in Table 2 the *distance metrics* do improve for 3D-Premise over Generated – so self repair is breaking some code so that it no longer compiles, but is possibly improving the distance metrics on the code that it does help on.
> > >
> > > Thank you for the responses and clarifications – I will maintain my rating.

---

### Official Review · Reviewer_Ghys · 2024-11-04

**Soundness:** 2
**Presentation:** 3
**Contribution:** 3
**Rating:** 6
**Confidence:** 4

**Summary:**

The paper introduces CADCodeVerify, a approach to iteratively verify and improve 3D objects generated from CAD code using Vision-Language Models (VLMs).  The method involves generating CAD scripting code from natural language prompts, executing the code to render a 3D object, and refining the code based on visual feedback through a question generation and answering process, to correct deviations from the initial specifications.  The approach is evaluated using CADPrompt, a benchmark dataset of 200 3D objects with corresponding natural language prompts and expert-annotated code.  The approach is evaluated on GPT-4, Gemini 1.5 Pro and CodeLLama models.

**Strengths:**

-	The paper comes with a new benchmark suite (CADPrompt) which is a well-curated, crowd-sourced benchmark with annotations and quality checks, which is a valuable resource for assessing CAD code generation and refinement methods.
-	CADCodeVerify uses an interesting novel idea to eliminate the need for human involvement by generating validation questions and answering them using VLMs.
-	The geometric solver-based baseline is very interesting and gives an upper estimate of the self-refinement process; it would be interesting to explore if this solver could also be used as a metric for evaluation.

**Weaknesses:**

-	The fundamental differences between CADCodeVerify and 3D-Premise are unclear. For example, it’s not specified whether 3D-Premise uses execution-error-based repair. Also, both approaches seem to use totally different prompts, so it is not clear if it is just a matter of better prompting or something fundamental (such as the question-answer based method)
-	The paper would be stronger if the approach was also evaluated on the 3D premise dataset
-	Some other details are missing (see below questions) regarding ambiguity in NL input and chain-of-thought based extension to 3D-Premise.

**Questions:**

1. Would providing images of the generated object from multiple viewpoints improve 3D-Premise’s performance? What about using a chain of thought prompting (for e.g. explicitly ask the model to reason whether each criterion in the initial task are satisfied)? This will be similar to CADCodeVerify, but doing 1 VLM call instead of doing 3 VLM calls in the verify step  to reduce the inference time.

2. In Figure 3, does 3D-Premise also repair code based on compile error messages?

3. How do you handle ambiguity in inputs in ground truth solutions? For example, if the NL input does not specify a particular size of hole, there could be many ground truth solutions which might impact the cloud-distance based metrics.

4. Since Point Cloud distance and Hausdorff distance are noisy metrics, how does the generated CAD program compare on the geometric metrics obtained using the geometric solver?

Minor:
- Consider renaming success rates to compile rates or something similar to convey that is only compilation success rate and not overall task success rate.

- For figures 5,6 (running examples), it will be useful to also see the repair steps.

---

> ### Author Response · Authors · 2024-11-21
> **Response to Reviewer Ghys**
>
> We thank this reviewer for their time and thorough review of our submission. We hope to address your concerns as follows:
>
> ## Does 3D-Premise also repair code based on compile error messages
>
> Yes, our implementation of 3D-Premise also leverages the code execution step presented in Section 3.2. For both the Geometric Solver and 3D-Premise baselines, all components of the CAD Code Generation process are kept consistent besides the method of generating feedback for code-refinement.
>
> ## Handling ambiguity of language-descriptions
>
> In our work, we adopted extensive review procedures to ensure that the descriptions are consistent with the target object as described in Section 4.1. In paragraph two of our limitations section, we note that the same object can be described in different ways depending on the specifier, and conversely, multiple object configurations may satisfy a single NL description. However, such incongruities are inherent to any approach combining freeform language with parametric 3D designs. This problem could be alleviated by providing K possible outputs for every description to compute a top-k metric, however, we leave that exploration to future work.
>
> ## Point-Cloud distance and Hausdorff distance are noisy
>
> To account for potential noise within the PC or Hausdorff distance measurement, we also included a third metric which evaluates overlap in the objects in the 3D-space (i.e., ntersection over ground truth (IoGT)) instead of computing a point-to-point measurement. All three of these metrics are well established measurements utilized in prior work to evaluate 3D Generations [1,2,3]. By showing consistent results across these three metrics, we believe that we have provided a reliable measurement of the competency of various VLMs and CADCodeVerify for the CADPrompt benchmark. If this reviewer believes that our paper would benefit from including the additional geometric-solver measurement proposed, we will compute that for the camera-ready version of the paper.
>
>
>
> [1] - Yuan, Zeqing, et al. "3D-PreMise: Can Large Language Models Generate 3D Shapes with Sharp Features and Parametric Control?." arXiv preprint arXiv:2401.06437 (2024).
>
> [2] - Sun, Yongbin, et al. "Pointgrow: Autoregressively learned point cloud generation with self-attention." Proceedings of the IEEE/CVF Winter Conference on Applications of Computer Vision. 2020.
>
> [3] - Vahdat, Arash, et al. "Lion: Latent point diffusion models for 3d shape generation." Advances in Neural Information Processing Systems 35 (2022): 10021-10039.
>
> ## Renaming Success-Rates
>
> We thank the reviewer for their suggestion, and we will replace “Success-Rate” with “Compile-rate” throughout the paper.

---

> > ### Author Response · Authors · 2024-11-25
> > **Response to Reviewer Ghys**
> >
> > Dear Ghys,
> >
> > Thank you for your valuable feedback and engagement with our paper. As the discussion period is coming to an end, we wanted to check if there are any unresolved concerns or additional clarifications you would like us to address. We are eager to ensure that all your questions are fully answered and that we can provide any further information to support your assessment.

---

> > > ### Comment · Reviewer_Ghys · 2024-11-26
> > > **Reply to authors**
> > >
> > > Thanks for rebuttal. However, it seems like not all of my questions were answered. For e.g. Would providing images of the generated object from multiple viewpoints improve 3D-Premise’s performance? What about using a chain of thought prompting (for e.g. explicitly ask the model to reason whether each criterion in the initial task are satisfied)? This will be similar to CADCodeVerify, but doing 1 VLM call instead of doing 3 VLM calls in the verify step to reduce the inference time.
> > >
> > > For ambiguity in inputs, my question was more if this affects the evaluation metrics (for e.g. the model could have found a solution is the correct with respect to the intent, but different from the ground truth object, and hence affecting the evaluation metrics). Also, from your manual reviews, how often do you find the intents ambiguous. For example, I see several instances from the examples in the paper (Figure 2) that use phrases like large rectangle (without a specific dimension) or slightly smaller (rather than specifying exactly how much smaller). If significant number of intents in your dataset are of this form, I think it greatly impacts the results and the conclusions of this paper (essentially making them not so useful).

---

> > > > ### Author Response · Authors · 2024-12-02
> > > > **Response to Reviewer Ghys**
> > > >
> > > > Dear Ghys,
> > > >
> > > > Thank you for pointing out that some of your concerns were not fully addressed in our initial response. We sincerely apologize for this oversight and greatly appreciate your detailed feedback. Below, we have provided thorough answers to the remaining questions you raised.
> > > >
> > > > ## 3D-Premise with Multiple Viewpoints and Chain-of-Thought Prompting:
> > > >
> > > > We conducted additional experiments on 3D-Premise, incorporating multiple viewpoints of images during the refinement process. Specifically, we included four images taken from different angles (0°, 90°, 180°, and 270°). Furthermore, we performed experiments using chain-of-thought prompting techniques for 3D-Premise. Additionally, we combined both multiple viewpoints and chain-of-thought prompting in our experiments.
> > > >
> > > > These experiments were conducted using GPT-4 in a few-shot setting. As demonstrated in the table below, our approach, CADCodeVerify, continues to outperform the new baselines.
> > > >
> > > > | **Feedback Mechanism**              | **IoGT ↑**          | **Point Cloud dist. ↓** | **Hausdorff dist. ↓**    | **Compile Rate ↑**  |
> > > > |--------------------------------------|---------------------|-------------------------|--------------------------|---------------------|
> > > > | **Generated**                        | 0.939 (0.030)       | 0.155 (0.140)           | 0.494 (0.368)            | 96.0%               |
> > > > | **3D-Premise**                       | 0.942 (0.033)       | 0.137 (0.155)           | 0.446 (0.396)            | 91.0%               |
> > > > | **3D-Premise - Multiple viewpoints** | 0.941 (0.035)       | 0.132 (0.147)           | 0.437 (0.395)            | 91.0%               |
> > > > | **3D-Premise - CoT**                 | 0.941 (0.031)       | 0.131 (0.142)           | 0.432 (0.409)            | 92.0%               |
> > > > | **3D-Premise - CoT & Multiple viewpoints** | 0.941 (0.031) | 0.150 (0.162)           | 0.477 (0.367)            | 90.0%               |
> > > > | **CADCodeVerify (Ours)**             | **0.944 (0.028)**   | **0.127 (0.135)**       | **0.419 (0.356)**        | **96.5%**           |
> > > > | **Geometric solver***                | **0.944 (0.031)**   | **0.103 (0.152)**       | **0.399 (0.433)**        | 95.5%               |
> > > >
> > > >
> > > > ## Ambiguity in inputs
> > > > We agree that ambiguous intents (e.g., phrases like “large rectangle” or “slightly smaller”) can lead the model to generate solutions that align with the intent but differ from the ground truth parameters, such as height, width, dimensions, or volume. This could limit the utility of evaluation metrics for comparing methods.
> > > >
> > > > To address this limitation, we propose a human evaluation protocol where outputs are manually rated on a scale from 0 to 100, focusing on their adherence to the intent rather than strict correspondence to the ground truth. This approach provides a more nuanced assessment of model performance and its practical applicability.
> > > >
> > > > As a proof of concept, we conducted a human evaluation on one experiment —specifically, the GPT-4 few-shot setting (200 objects), which demonstrates the highest performance in Table 2 of the paper. Due to time constraints during the rebuttal period, one of the authors conducted this evaluation. Future work includes human evaluations, with 30 participants each assessing 15 generated objects, the results of this experiment will be incorporated into a camera-ready version
> > > >
> > > >
> > > > The evaluation methodology was as follows:
> > > >
> > > > 1. The evaluator read the natural language description of the 3D object (input).
> > > >
> > > > 2. For each refinement approach—3D-Premise, Geometric Solver, and CADCodeVerify—the evaluator assigned a score from 0 to 100 based on the following criteria:
> > > >
> > > > - 0: The refined code did not compile or failed to render a 3D object.
> > > >
> > > > - 10: There was a logic error, resulting in implausible configurations of 3D objects that did not resemble real-world contexts.
> > > >
> > > > - 10–50: The refined 3D object differed from the input description.
> > > >
> > > > - 50–75: The refined 3D object was slightly different from the input description but exhibited some similarities.
> > > >
> > > > - 75–90: The refined 3D object was mostly similar to the input description, with minor differences.
> > > >
> > > > - 90–100: The refined 3D object matched the input description exactly.
> > > >
> > > >
> > > >
> > > > The table below summarizes the human evaluation results for each refinement approach, reporting the mean, median, standard deviation (SD), minimum, and maximum scores.
> > > >
> > > > Feedback Mechanism | Mean   | Median | SD    | Minimum | Maximum
> > > > -------------------|--------|--------|-------|---------|---------
> > > > **3D-Premise**         | 68.95  | 90.00  | 35.79 | 0.00    | 100.00
> > > > **Geometric solver**   | 67.52  | 80.00  | 34.91 | 0.00    | 100.00
> > > > **CADCodeVerify**      | 74.52  | 90.00  | 32.64 | 0.00    | 100.00
> > > >
> > > > We hope these experiments address the reviewer’s concerns. Please feel free to share any additional feedback or questions—we would be glad to discuss further and work towards enhancing your evaluation of our work.

---

### Official Review · Reviewer_jRaW · 2024-11-05

**Soundness:** 3
**Presentation:** 2
**Contribution:** 3
**Rating:** 6
**Confidence:** 3

**Summary:**

The paper entitled “GENERATING CAD CODE WITH VISION-LANGUAGE MODELS FOR 3D DESIGNS” proposes an approach to generate and verify CAD 3D objects from natural language. The proposed approach introduces a verification step called CADCodeVerify. CADCodeVerify use Vison Language Machine (VLM) to verify CAD objects against generated visual questions. The hypothesis is that the question serves as visual properties the object should satisfy to meet the structural specification. The key contribution of this approach is the use of a refinement step. From the language prompt that describes the object in natural language, a first CAD description code (CADQuery) is generated. The visual questions are also generated from this language prompt input. The object (CADQuery) is verified against visual properties questions using a VLM that generates feedback for each questions. This feedback is used to refined the CADQuery code.

To summarise, the contributions of this work is two fold: 1) an automated CAD generation approach that does not necessitate humain interaction; 2) a dataset called CADPrompt that comprises 200 CAD objects. Expermental results shows when implemented using GPT-4 LLM, we observe improvements in object generation accuracy and an increased success rate compared to other LLMs.

**Strengths:**

- Novelty and Originality: The integration of question/answering based VLM to refine the object quality.

- Evaluation: The increase of success rate of compilable outputs at the end of the refinement shows that the method can improve the generation of CAD code from language prompt.

- Soundness: The technical approach is sound and could be applied to a lot of CAD or CAV tasks. The approach is a specification refinement method using generative AI. From an initial specification, an initial object plus query are generated. Then, feedback from query satisfaction are used to improve the object. A dataset is provided and metrics are determined to rank the quality of the generated object. This has applications beyond CAD specifically.

**Weaknesses:**

- Dataset Scope: CADPrompt could have been better introduce, showing the most complex objects in terms of complexity and difficulty metrics.

- Complexity Metric for Objects: This paper measures object complexity by counting vertices and faces. However, using metrics like bounding boxes or decomposed bounding volumes could better reflect structural complexity. Vertices mainly define shape details, not true complexity—a simple shape like a cube can have many polygons, while complex shapes like an aircraft wing might need fewer. In physics engines, object complexity is often defined by the volume structure the object requires. Lower polygons can shape more complex structures.

- Difficulty Metric for Objects: The current method of measuring complexity through word and line count for the natural language descriptions and Python code may not fully capture their complexity. A semantic-based complexity metric, where each instruction has a complexity value, would better reflect how nuanced or contextually challenging the description and code are, which could impact model performance.

- Clarity and Readability: Certain sections are overly technical and hard to understand without referencing the appendix. Simplifying these sections would improve accessibility for readers. Terminological inconsistencies between LLM used for language to code and code with feedback to code and VLM for image and questions to answers also introduce potential confusion, which could be resolved for clarity. Certain sections are overly technical and hard to understand without referencing the appendix. Simplifying these sections could enhance accessibility for readers. For example, consider moving equations 1-7, which aren't essential for understanding the main contributions, to the appendix. Conversely, evaluation metric equations (8-10), such as those for point cloud distances, Hausdorff distance, and IoGT, should remain in the main text. Additionally, bringing valuable visuals currently in the appendix (Figures 5, 11, 12, and 14) into the main text would ease the understanding and improve work illustration. The paper would also benefit from a stronger focus on scientific insights and hypothesis testing over too much detailled descriptions. For instance, providing context around why a fixed number of five verification questions is used, without adapting to object complexity, would clarify the rationale. Similarly, explaining why only two refinement iterations are applied, regardless of complexity, would better illustrate the scientific intuitions driving these choices. Furthermore, details on how AI context like 7, 9, and 10, have been engineered such as crafted based on expert knowledge or through authors tuning would be valuable. Shifting focus to the "why" rather than the "what" would help readers grasp the paper challenges.

- Positioning: the paper’s positioning within the state of the art could be made clearer, with more explicit distinctions from existing work. The paper integrates the related work 3D-premise in the experiments, but didn't explicitly state the approach difference. The feedback is based on question answering rather than the initial description of the object.

**Questions:**

- Q1: I appreciate the framework’s contribution, but could you provide a bit more on how this specifically differs from or builds upon prior work (3D-Premise)? Is it an improvement or an entire new approach?

- Q2: Can you tell us more about the quality and diversity of the CADPrompt dataset? I’m curious about how it’s tailored to the goals of your approach. A bit more on why it’s a good benchmark—perhaps a breakdown of object types, complexity levels, or any specific challenges it presents?

- Q3: The experimental results look interesting, but it would be good to understand more in details where the approach performs well and where it might have limitations. Do you have any insight to share about this?

---

> ### Author Response · Authors · 2024-11-21
> **Response to Reviewer jRaW Part(1/2)**
>
> We thank this reviewer for their measured and insightful feedback. We hope our response will sufficiently address the concerns raised.
>
> ## Compilation Difficulty Metric
> It appears that the reviewer may have misunderstood the difficulty metric employed in our paper. Our “difficulty” metric defined in Section 4.3 is not computed as the “word and line count for the natural language descriptions and Python code” as stated by this reviewer. Instead, our difficulty metric is “measure of how difficult it is for a set of three language models (i.e., GPT-4, Gemini, and CodeLlama) to generate code for a given 3D object across two prompting methods (i.e., zero- and few-shot prompting), for a total of six attempts to generate compilable code” as stated in Section 4.3. This metric is one of three metrics we use for analyzing the models’ performances.
>
> ## Complexity metric for Objects (Mesh Complexity)
> In 3D-modeling software such as Blender or Unity, meshes with a higher polygon count allow for rendering objects in higher levels of detail, implying greater expressivity. While there may be objects with a lower number of faces or vertices which are semantically more complex or more difficult to manufacture, measuring the number of faces and vertices provides insight into the level of detail expressed in the 3D-object.
>
> ## Semantic Complexity metric
>
> To better understand the challenge proposed by the objects in CADPrompt, we conducted a semantic evaluation as suggested by the reviewer. We recruited an independent mechanical engineer to semantically rate the complexity of 200 examples in our dataset. The annotator utilized a four-point scale as follows:
>
> - (Simple) - The object is extremely basic, with few features. It may
> consist of one geometric shape.
>
> - (Moderately Complex) - The object has a moderate amount of detail,
> with a few distinct features or components
>
> - (Complex) - The object is complex, with many interconnected parts,
> fine details, or intricate shapes.
>
> - (Very Complex) - The object is highly intricate, with many
> components, detailed textures, or complex shapes. It may have a large
> number of fine details, interlocking parts, or unique geometric features
>
> The breakdown of our dataset according to this independent reviewer’s evaluation was as follows: Simple - 17, Moderate Complexity - 39, Complex - 87, and Very Complex - 57. When comparing the semantic complexity with our quantitative metrics, we observe that all our measures of “complexity” are generally aligned with this semantic complexity measurement (Please refer to Table 1 in the updated version.).
>
> ## Q1 – Differences from 3D-Premise
>
> 3D-Premise presented the first exploration of leveraging the vision-capabilities of VLMs for CAD Code Refinement. CADCodeVerify offers three key advancements over 3D-Premise.
>
> 1. Our approach adopts a “Self-Verification” method wherein the VLM is prompted to generate its own set of validation questions rather than answering a fixed set of questions decided by an expert, thereby removing the human-in-the-loop expertise.
>
> 2. 3D-Premise does not include the code execution stage (Section 3.2), which is crucial for reducing the number of syntax errors in the code.
>
> 3. 3D-Premise only offers qualitative insights regarding the performance of their proposed method to integrate visual feedback into CAD code refinement.  In our approach, we conduct a comprehensive and systematic evaluation of CADCodeVerify's capabilities using our CADPrompt benchmark, establishing a foundation for standardized research on CAD code generation.
>
> ## Q3 - More details regarding where the approach performs well and its limitations.
> We conducted human-in-the-loop experiments to compare its performance with CADCodeVerify. Human feedback was used to provide precise instructions on the exact changes needed for the 3D object. The results demonstrate a slight improvement compared to CADCodeVerify in terms of Point Cloud distance and Hausdorff distance, with performance metrics improving from 0.137 and 0.445 to 0.120 and 0.397, respectively (See Table 4). These findings suggest that CADCodeVerify delivers feedback that closely resembles gold-standard feedback.
>
> Some 3D objects are highly complex, making it challenging for LLMs to generate them. Additionally, the feedback provided by both CADCodeVerify and human reviewers is often insufficient to refine these objects due to their complexity.  Notably, CADCodeVerify demonstrates a significant improvement in correcting 3D objects with structural errors (see Figure 8). In future work, we plan to fine-tune LLMs for domain-specific tasks in CAD code generation, aiming to enhance the quality of CAD outputs significantly.

---

> > ### Author Response · Authors · 2024-11-21
> > **Response to Reviewer jRaW Part(2/2)**
> >
> > ## Clarity/Readability
> > We thank the reviewer for their suggestions regarding improving the clarity/readability of our paper. We have updated the text in the methodology/methods section of the paper to improve clarity/readability. While we were unable to move Figure 9, of the appendix, to the main paper due to space constraints, we have added an abridged version of this figure to the main paper (Figure 5, in the updated version of the paper).
> >
> > Regarding the equations Section 3, we chose to utilize equations in our description to provide a concise and modular explanation of each component of our approach in a consistent manner to prior work on code generation and refinement.  If the reviewer strongly feels as though the equations should not be in the main paper, we can move them to the appendix in the camera-ready version
> >
> > ## Justification for choices in methodology
> >
> > - Why were two refinement steps used? -- This information can be found in Section B.3 in the Appendix. We did not observe any improvement after the second stage of refinement. Therefore, we restricted the number of refinement steps to two. This result is consistent with prior works on code-refinement which also showed that the effectiveness of code-refinement wanes after two iterations of feedback [1,2]
> >
> > - Why do we use a five verification questions without considering object complexity? -- We do not use a fixed number of questions per object. As stated in the “Question-Answer Generation” subsection in Section 3.3, “CADCodeVerify generates between two to five questions per example.” Therefore, we prompt the model to generate as many questions as it deems appropriate, with five selected as a reasonable upper bound to constrain the outputs. We can add additional examples of when CADCodeVerify generates a different number of questions (i.e., not five) to the appendix if the reviewer would like to see those.
> >
> > [1] - Madaan, Aman, et al. "Self-refine: Iterative refinement with self-feedback." Advances in Neural Information Processing Systems 36 (2024).
> >
> > [2] - Chen, Hailin, et al. "Personalised distillation: Empowering open-sourced llms with adaptive learning for code generation." arXiv preprint arXiv:2310.18628 (2023).

---

> ### Author Response · Authors · 2024-11-25
> **Response to Reviewer jRaW**
>
> Dear jRaW,
>
> We hope you have had a chance to review our rebuttal. As today is the last day of the discussion period, we wanted to check-in to see if there was anything you wanted to discuss with us regarding our paper. Please let us know if there are any outstanding concerns that you would like us to address or discuss towards improving your assessment of our paper.

---

> > ### Comment · Reviewer_jRaW · 2024-11-27
> >
> > Dear authors,
> >
> > Thank you for your rebuttal and the clarifications. I am satisfied with the answers about compilation difficulty and semantic complexity metrics. Generally, I remain positive that the paper has value. I have follow-up comments/questions on the other answers. Time might lack to address them but you might want to consider them in potential revisions of your paper.
> >
> > **Object/mesh complexity**
> >
> > Once again, higher polygon counts do not necessarily indicate greater complexity, nor does lower complexity guarantee fewer polygons. For instance, spherical objects often have higher polygon counts but are not structurally complex… Anyway, are you suggesting that the cadquery commands used in the generated code aim to avoid unnecessary face subdivisions or excessive polygons generation while accurately capturing the geometry of the objects?
> >
> > **Key advances over 3D-premise**
> >
> > 3D-Premise does not require expert-defined questions and refines fully automatically, using the image input along with the (simplified) instruction “Does the object correspond to the image” for feedback. It also utilizes Blender3D for code execution but does not include a code repair stage. Are you referring to a “...code repair stage to reduce syntax errors”?
> >
> > Still, the positioning could be more clear and explicit also because it is a baseline that you implemented yourself.
> >
> > **Performance Insights and Limitations**
> >
> > Thank you for this response. While "excels" might be too strong a term, it appears that our proposed question-based refinement has enhanced the LLMs' performance in these tasks. However, I recommend exploring a more collaborative human/machine approach in future work to facilitate the application of such support in real-world and industrial contexts.
> >
> > **Methodology Justifications**
> >
> > Could you elaborate on the rationale behind these observations, specifically the notion of diminishing returns after two iterations and the importance of bounding questions to maintain efficiency? It would be helpful to understand the underlying reasoning or empirical evidence supporting these approaches, as well as how they contribute to optimizing the overall process.

---

> ### Author Response · Authors · 2024-12-01
> **Response to Reviewer jRaW Part (1/2)**
>
> Thank you for following up with your questions. We greatly appreciate your valuable feedback and the time you’ve taken to engage with us. We have addressed your concerns and are committed to ensuring that all your questions are fully answered. Please let us know if there’s any additional information we can provide to support your assessment.
>
> ## Object/mesh complexity
>
> Thank you for your follow-up question regarding mesh complexity. We agree with the reviewer that classifying 3D objects as simply "simple" or "complex" is imperfect. This limitation is evident in our dataset. For instance, while a circle is a simple shape conceptually, it consists of 73 vertices and 140 faces. In response, we have decided to move the discussion of mesh complexity in Section 4.3 to the Appendix.
>
> To address the reviewer’s concern, we generated an additional way to classify the complexity as “semantic complexity”. Following the reviewer's suggestion, we recruited an independent mechanical engineer to semantically rate the complexity of 200 objects, categorizing them as Simple, Moderately Complex, Complex, or Very Complex. We found the semantic complexity measure to be highly effective, as demonstrated by the dataset split based on semantic complexity shown in Table 8 of the Appendix. Further details on semantic complexity are provided in our previous response and in the paper on line 288.
>
>
>
> ## Key advances over 3D-premise
>
> Thank you for bringing this to our attention. We have incorporated a code repair stage into the 3D-Premise approach, and all three refinement approaches share the same code repair stage, as outlined in Equation 2.
>
> Furthermore, since 3D-Premise serves as our baseline, we conducted additional experiments to enhance its performance, including:
>
> - Incorporating multiple viewpoints by utilizing four images captured from different angles (0°, 90°, 180°, and 270°).
>
> - Applying chain-of-thought prompting within the 3D-Premise framework.
>
> The results of these experiments are provided in the table below for the GPT-4 few-shot setting:
>
> | **Feedback Mechanism**              | **IoGT ↑**          | **Point Cloud dist. ↓** | **Hausdorff dist. ↓**    | **Compile Rate ↑**  |
> |--------------------------------------|---------------------|-------------------------|--------------------------|---------------------|
> | **Generated**                        | 0.939 (0.030)       | 0.155 (0.140)           | 0.494 (0.368)            | 96.0%               |
> | **3D-Premise**                       | 0.942 (0.033)       | 0.137 (0.155)           | 0.446 (0.396)            | 91.0%               |
> | **3D-Premise - Multiple viewpoints** | 0.941 (0.035)       | 0.132 (0.147)           | 0.437 (0.395)            | 91.0%               |
> | **3D-Premise - CoT**                 | 0.941 (0.031)       | 0.131 (0.142)           | 0.432 (0.409)            | 92.0%               |
> | **3D-Premise - CoT & Multiple viewpoints** | 0.941 (0.031) | 0.150 (0.162)           | 0.477 (0.367)            | 90.0%               |
> | **CADCodeVerify (Ours)**             | **0.944 (0.028)**   | **0.127 (0.135)**       | **0.419 (0.356)**        | **96.5%**           |
> | **Geometric solver***                | **0.944 (0.031)**   | **0.103 (0.152)**       | **0.399 (0.433)**        | 95.5%               |

---

> > ### Author Response · Authors · 2024-12-01
> > **Response to Reviewer jRaW Part (2/2)**
> >
> > ## Performance Insights and Limitations
> >
> > Thank you very much for your valuable feedback. We agree that your recommendation to explore a collaborative human-machine approach is an intriguing idea, and we plan to pursue it in a follow-up paper. For this rebuttal, we conducted preliminary experiments as a proof of concept to demonstrate how humans and machines can collaborate to provide feedback. These experiments were compared against two baselines: (1) the human-in-the-loop approach (Human) and (2) our proposed approach, CADCodeVerify (Machine). The details of these experiments are as follows:
> >
> >
> > The results of these experiments are presented below, demonstrating that the collaborative human-machine approach outperforms both baselines for the GPT-4 few-shot setting:
> >
> >
> >
> >
> > - **Human-in-the-loop (Human):** In this experiment, a human (the author) directly provides feedback to LLMs to refine the generated object.
> >
> > - **CADCodeVerify (Machine):** Our proposed approach, CADCodeVerify, delivers feedback to the LLMs automatically without human intervention.
> >
> > - **Collaborative human-machine (Human and Machine):**  The machine first generates questions based on language prompts and creates four images of the generated object from different angles (0°, 90°, 180°, 270°). The human selects the best viewing angle, and the machine answers the questions using the chosen image. The human then verifies and corrects the machine's responses. Based on these verified answers, the machine generates feedback,  and also human adds additional feedback. The combined feedback from both the human and machine is used to refine the 3D object through iterative improvements.
> >
> > | **Feedback Mechanism**              | **IoGT ↑**          | **Point Cloud dist. ↓** | **Hausdorff dist. ↓**    | **Compile Rate ↑**  |
> > |--------------------------------------|---------------------|-------------------------|--------------------------|---------------------|
> > | **Generated**                        | 0.930 (0.043)       | 0.156 (0.138)           | 0.495 (0.287)            | 98.5%               |
> > | **CADCodeVerify (Machine)**          | **0.948 (0.036)**   | 0.137 (0.136)           | 0.445 (0.302)            | 98.5%               |
> > | **Human-in-the-Loop (Human)**        | 0.944 (0.032)       | 0.120 (0.140)           | 0.397 (0.354)            | **99.0%**           |
> > | **Collaborative Human and Machine**  | 0.947 (0.025)       | **0.087 (0.178)**       | **0.33 (0.552)**         | **99.0%**           |
> >
> >
> > ## Methodology Justifications
> >
> > ### **Rationale for limiting to two Iterations:**
> >
> > As noted in Table 5, 6 and 7 (in the Appendix), the iterative refinement process with CADCodeVerify demonstrated no significant improvement in metrics like Point Cloud Distance and Compile Rate, Hausdorff Distance beyond two iterations. This aligns with prior studies in iterative code refinement (e.g., Madaan et al., 2023), suggesting that diminishing returns after two iterations are common in such scenarios.
> >
> > Empirically, we report the results for both refine-1 and refine-2 across all approaches and three LLMs: GPT-4 (Table 5), Gemini (Table 6), and CodeLlama (Table 7). In many cases, refine-2 performs worse than refine-1 or shows only a slight improvement.
> >
> >
> >
> > ### **Importance of Bounding Questions:**
> >
> > Bounding the number of validation questions between two to five ensures that the refinement process is directed towards addressing the most critical discrepancies in the generated designs. This avoids overloading the refinement loop with excessive or irrelevant feedback, which could dilute its effectiveness.
> >
> >
> >
> > Empirically, we randomly selected 100 examples and conducted two experiments using GPT-4. The details of these experiments are as follows:
> >
> > - **Zero-shot QA generation:** In this experiment, questions were generated in a zero-shot manner without providing any demonstration examples or restricting the number of questions.
> >
> > - **CADCodeVerify:** This is our proposed approach, where few-shot examples were provided to the LLMs, along with instructions to generate between two and five questions.
> >
> > The results, as shown in the table below, indicate that LLMs perform better when questions are generated using few-shot demonstrations and the number of questions is limited.
> >
> > | **Ablation Study**          | **IoGT ↑**          | **PLC distances ↓**       | **Hausdorff dist. ↓**     | **Compile Rate ↑**  |
> > |-----------------------------|---------------------|---------------------------|---------------------------|---------------------|
> > | **Generated**               | 0.909 (0.062)      | 0.156 (0.150)             | 0.491 (0.348)             | 96.5%               |
> > | **Zero-shot QA generation** | **0.919 (0.049)**  | 0.141 (0.112)             | 0.471 (0.280)             | **98.0%**           |
> > | **CADCodeVerify (Ours)**    | **0.919 (0.045)**  | **0.126 (0.122)**         | **0.444 (0.308)**         | 97.5%               |

---

> > > ### Author Response · Authors · 2024-12-01
> > > **Response to Reviewer jRaW**
> > >
> > > Please feel free to share any additional feedback or questions—we are happy to discuss further and work toward improving your evaluation of our paper.

---

### Meta-Review · Area_Chair_3AfN · 2024-12-18

**Metareview:**

In this paper, the authors introduce CADCodeVerify, a novel approach that uses VLMs for iterative verification and refinement of CAD code generation. To evaluate CADCodeVerify, the authors also propose CADPrompt, a benchmark for CAD code generation consisting of 200 natural language prompts paired with expert-annotated scripting code for 3D objects. Under this benchmark, the experiments include comparisons with existing methods as well as with human experts, demonstrating that the proposed method is superior.

However, there are a limited number of technical contributions. The main contribution seems to be the novel mechanism of using off-the-shelf LLMs and VLMs to automatically generate high-quality CAD code. Though the experiments show superior results, the small scale of the experiments makes it difficult to conclude that the pipeline is universally applicable. In addition, the method's reliance on natural language input may not account for precise numerical specifications, which is another limitation. Therefore, I recommend accepting the paper but as a poster.

**Additional Comments On Reviewer Discussion:**

The reviewers primarily expressed concerns about the following points:
1. The initial submission did not provide sufficient comparisons with existing methods in the field.
2. The distinctions between the proposed method and existing approaches, particularly 3D-Premise, were not clearly articulated.
3. Important details of the proposed method were not adequately explained.
4. The nature of the natural language input used by the method was ambiguous.
5. The size of the dataset is small.
6. Some design choices in the proposed method appeared to be empirical without sufficient justification (e.g., reasons why the results are superior to those of other methods, the number of refinements).

During the rebuttal phase, concerns 1, 2, and 3 were satisfactorily addressed by the authors. However, points 4, 5, and 6 remain as limitations of the work. Despite these issues, the reviewers agree that the results presented are positive and show promise.

---

### Decision · Program_Chairs · 2025-01-22

Accept (Poster)